# Activation of mTORC1 and c-Jun by Prohibitin1 loss in Schwann cells may link mitochondrial dysfunction to demyelination

**Gustavo Della-Flora Nunes[1,2]\*, Emma R Wilson[1,2]\*, Edward Hurley[1]\*, Bin He[3], Bert W O'Malley[4]\*, Yannick Poitelon[5]\*, Lawrence Wrabetz[1,2,6]\*, M Laura Feltri[1,2,6]\***

[1]Hunter James Kelly Research Institute, University at Buffalo, Buffalo, United States; [2]Department of Biochemistry, University at Buffalo, Buffalo, United States; [3]Immunobiology & Transplant Science Center and Department of Surgery, Houston Methodist Hospital, Houston, United States; [4]Department of Medicine and Molecular and Cellular Biology, Baylor College of Medicine, Houston, United States; [5]Department of Neuroscience and Experimental Therapeutics, Albany Medical College, Albany, United States; [6]Department of Neurology, Jacobs School of Medicine and Biomedical Sciences, University at Buffalo, Buffalo, United States

**\*For correspondence:**
gnunes@buffalo.edu (GD-FN);
ewilson5@buffalo.edu (ERW);
edwardhu@buffalo.edu (EH);
berto@bcm.edu (BWO'M);
poitely@amc.edu (YP);
lwrabetz@buffalo.edu (LW);
mlfeltri@buffalo.edu (MLauraF)

**Competing interest:** The authors declare that no competing interests exist.

**Abstract** Schwann cell (SC) mitochondria are quickly emerging as an important regulator of myelin maintenance in the peripheral nervous system (PNS). However, the mechanisms underlying demyelination in the context of mitochondrial dysfunction in the PNS are incompletely understood. We recently showed that conditional ablation of the mitochondrial protein Prohibitin 1 (PHB1) in SCs causes a severe and fast progressing demyelinating peripheral neuropathy in mice, but the mechanism that causes failure of myelin maintenance remained unknown. Here, we report that mTORC1 and c-Jun are continuously activated in the absence of *Phb1*, likely as part of the SC response to mitochondrial damage. Moreover, we demonstrate that these pathways are involved in the demyelination process, and that inhibition of mTORC1 using rapamycin partially rescues the demyelinating pathology. Therefore, we propose that mTORC1 and c-Jun may play a critical role as executioners of demyelination in the context of perturbations to SC mitochondria.

## Introduction

Schwann cells (SCs) are the main glial cell type of the peripheral nerves, where they closely associate with axons (for review, see *Wilson et al., 2021*). Many axons extend very far from neuronal cell bodies, preventing fast delivery of essential cellular substrates. Therefore, SCs are believed to provide essential trophic and metabolic support to nearby axons (*Nave, 2010*). In addition, SCs identify axons larger than 1 μm in diameter and wrap them in multiple layers of a specialized membrane extension known as myelin. The myelin sheath reduces the capacitance of the axonal membrane, and its discontinuous structure enables 'saltatory conduction', whereby ionic exchanges are concentrated in small myelin-free regions called nodes of Ranvier. The importance of SCs is evident from the great variety of inherited and acquired peripheral neuropathies that arises when these cells are impaired (*England and Asbury, 2004*).

Myelin is intuitively perceived as a stable structure, which can be exemplified by the remarkable discovery of preserved myelin ultrastructure in a 5000-year-old ice man (*Hess et al., 1998*). This notion of stability was initially confirmed by studies investigating the turnover of myelin components

in brain, with many myelin proteins and lipids showing half-lives of months (*Smith and Eng, 1965*; *Fischer and Morell, 1974*). Nevertheless, even early studies were quick to point out that some of the myelin components had much faster turnover rates (*Singh and Jungalwala, 1979*; *Sabri et al., 1974*; *Hayes and Jungalwala, 1976*), suggesting that portions of the myelin (especially its non-compact regions) could be more dynamic. Furthermore, we now know that maintenance of the myelin structure is not passive, requiring sustained expression of the transcription factors EGR2 (also known as Krox20) (*Decker et al., 2006*) and SOX10 (*Bremer et al., 2011*) in SCs, as well as continuous synthesis of myelin proteins (*Meschkat et al., 2020*) and lipids (*Zhou et al., 2020*) in brain.

Recently, mitochondria and cell metabolism were also implicated in myelin formation and maintenance in SCs. We reported that ablation of the primarily mitochondrial protein prohibitin 1 (*Phb1*) in SCs greatly impairs myelin maintenance in the PNS (*Della-Flora Nunes et al., 2021*). In addition, SC-specific deletion of the mitochondrial transcription factor *Tfam* (*Viader et al., 2011*), the respiratory chain component *Cox10* (*Fünfschilling et al., 2012*), the metabolic regulator *Lkb1* (*Beirowski et al., 2014*; *Pooya et al., 2014*; *Shen et al., 2014*), the nicotinamide mononucleotide synthetizing enzyme *Nampt* (*Sasaki et al., 2018*), or the nutrient-sensing O-linked N-acetylglucosamine transferase *Ogt* (*Kim et al., 2016*), all lead to peripheral neuropathy phenotypes in mice. Nonetheless, the mechanisms linking mitochondrial dysfunction to impaired myelin maintenance in SCs remain poorly understood.

Two interesting candidate pathways that may be activated in the context of mitochondrial dysfunction in SCs are mTORC1 and c-Jun. The mechanistic target of rapamycin (mTOR) is a serine/threonine kinase that regulates cellular growth and catabolism/anabolism rate according to the availability of nutrients and other cellular resources (*Saxton and Sabatini, 2017*). Functionally, mTOR exerts its activity in protein complexes known as mTOR complex 1 (mTORC1) and mTORC2, with mTORC1 being more widely studied. In SCs, mTORC1 plays a dual role: its high activity during development regulates SC proliferation and prevents premature differentiation, while its low but continuous activity in myelinating SCs drives myelin sheath growth (*Beirowski et al., 2017*; *Figlia et al., 2017*; *Jiang et al., 2018*). mTORC1 phosphorylates several substrates, among which are the extensively studied 4E-BP1 and ribosomal protein S6 kinase (S6K). S6K, in turn phosphorylates many other proteins, the first identified being the 40 S ribosomal protein S6. Through phosphorylation of its targets, mTORC1 exerts many functions (for review, see *Saxton and Sabatini, 2017*), including: 1. Regulation of global translation levels; 2. Modulation of the metabolism of glucose, glutamine and lipids; 3. Suppression of autophagy; 4. Control of cell size. In SCs, the specific downstream targets of mTORC1 have seldomly been explored, but S6K seems to mediate the developmental suppression of EGR2, preventing precocious SC differentiation (*Figlia et al., 2017*). mTORC1 is also temporarily activated after nerve injury to induce expression of c-Jun (an AP-1 transcription factor) (*Norrmén et al., 2018*). c-Jun is the master transcription factor orchestrating the transdifferentiation of SCs to a repair phenotype that mounts a response favoring axon regrowth, tissue reinnervation and clearance of myelin debris (*Jessen and Mirsky, 2019*). In the case of myelinating SCs, this transdifferentiation process involves myelin removal followed by its autophagic degradation (myelinophagy) (*Gomez-Sanchez et al., 2015*). This, associated to the fact that enforced *Jun* expression is sufficient to trigger demyelination (*Fazal et al., 2017*), raises the possibility that c-Jun and mTORC1 may play a role in the dismantling of myelin in the context of peripheral neuropathies.

Here we report that mice lacking *Phb1* in SCs (Phb1-SCKO) activate a response involving c-Jun and the mTORC1 pathway, possibly directly downstream of the mitochondrial damage. In addition, both c-Jun and mTORC1 seem to participate in the demyelination process in Phb1-SCKO animals. Moreover, inhibition of mTORC1 using rapamycin is able to partially rescue morphological and functional aspects of the phenotype of Phb1-SCKO mice. These results reveal a previously unknown mechanism that contributes to myelin loss secondary to SC mitochondrial damage. Furthermore, our findings implicate c-Jun and mTORC1 in the SC adaptation to mitochondrial dysfunction, raising the possibility that c-Jun and mTORC1 may also be involved in the response to mitochondrial damage in other cell types.

## Results

### Deletion of *Phb1* in SCs elicits fast and widespread myelin loss in the PNS

We recently reported that deletion of *Phb1* specifically in SCs causes a severe and fast progressing demyelinating phenotype in mice (*Della-Flora Nunes et al., 2021*). Mice lacking *Phb1* in SCs (Phb1-SCKO) show the first signs of myelin loss at postnatal day 20 (P20), with demyelination peaking around P40-P60 and causing partial or complete hindlimb paralysis around P90 (*Della-Flora Nunes et al., 2021*). Concomitant with the demyelination, Phb1-SCKO mice also display axonal degeneration, which follows a similar temporal progression (*Della-Flora Nunes et al., 2021*). To extend this data, we report here similar findings from density analysis in electron micrographs from the same animals (*Figure 1A and B*). This pathology is likely caused by the profuse changes in mitochondrial morphology and function that follow deletion of *Phb1* in SCs (*Della-Flora Nunes et al., 2021*). We also demonstrated that SC apoptotic cell death is not greatly altered by deletion of *Phb1* in vivo, and that SC numbers remain normal in nerves of Phb1-SCKO, suggesting that a simple loss of viability cannot explain their pathology (*Della-Flora Nunes et al., 2021*). Nevertheless, the molecular mechanism linking mitochondrial damage to myelin destruction in Phb1-SCKO mice remains unknown.

### Ablation of *Phb1* in SCs leads to upregulation of c-Jun and activation of the mTORC1 pathway

Although little is known about how myelin is maintained long-term, it seems that preservation of myelin with the correct structure and thickness is an active process, requiring the continuous activation of certain cellular machinery in SCs (*Bremer et al., 2010*; *Bremer et al., 2011*; *Decker et al., 2006*). We therefore hypothesized that the SC response to mitochondrial dysfunction may inadvertently interfere with pathways critical for myelin maintenance. Seeking to find the molecular link between compromised mitochondrial function and demyelination, we investigated the status of the following molecular pathways previously reported to play roles in preservation of myelin: mTORC1 (important for myelination *Beirowski et al., 2017*; *Figlia et al., 2017*) and remyelination (*Norrmén et al., 2018*), ERK 1/2 (whose activation is sufficient to trigger demyelination *Napoli et al., 2012*), AKT (important to regulate myelin sheath thickness *Domènech-Estévez et al., 2016*), c-Jun (the master transcription factor of nerve repair *Arthur-Farraj et al., 2012*), and eIF2α (the core protein in the integrated stress response, ISR, which is important for myelin maintenance in the context of perturbed protein homeostasis in SCs *Scapin et al., 2020*; *D'Antonio et al., 2013*).

We had previously reported that phosphorylation of eIF2α is actually protective in Phb1-SCKO mice. In addition, Phb1-SCKO mice only show minor changes in the ERK 1/2 pathway, which are probably insufficient to initiate demyelination (*Della-Flora Nunes et al., 2021*). The same is true for the AKT pathway (*Figure 1—figure supplement 1*). On the other hand, Phb1-SCKO animals show continuous upregulation of c-Jun and activation of the mTORC1 pathway, as measured by the protein and phosphorylation levels of the downstream targets 40 S ribosomal protein S6 (S6) and 4E-BP1 (*Figure 1C and D* and *Figure 1—figure supplement 2*). Importantly, these changes are evident even before overt demyelination. Nonetheless, we found no differences in the phosphorylation levels of mTOR at Ser2448 (*Figure 1—figure supplement 1*), suggesting that the AKT/PI3K pathway is not involved in activating the mTORC1 pathway in Phb1-SCKO mice. Interestingly, the changes we detected involve upregulation of both total and phosphorylated levels of 4E-BP1 and S6, while the ratio of phosphorylated to total levels of these proteins remained mostly unaltered (*Figure 1—figure supplement 2* and *Figure 1—figure supplement 3*). Although we cannot completely rule out the contribution of other nerve cells to the levels of c-Jun, 4E-BP1 and S6, infiltrating macrophages do not seem to activate c-Jun and mTORC1 in sciatic nerves of Phb1-SCKO mice (*Figure 1—figure supplement 4*), suggesting that these alterations may primarily occur in SCs.

### mTORC1 and c-Jun can be activated downstream of mitochondrial perturbations in SCs

Both c-Jun and mTORC1 have been previously implicated in the cellular response to mitochondrial dysfunction. The mitochondrial unfolded protein response (mtUPR) is believed to be partly regulated by binding of an AP-1 transcription factor (postulated to be c-Jun) to the promoter of CHOP and C/

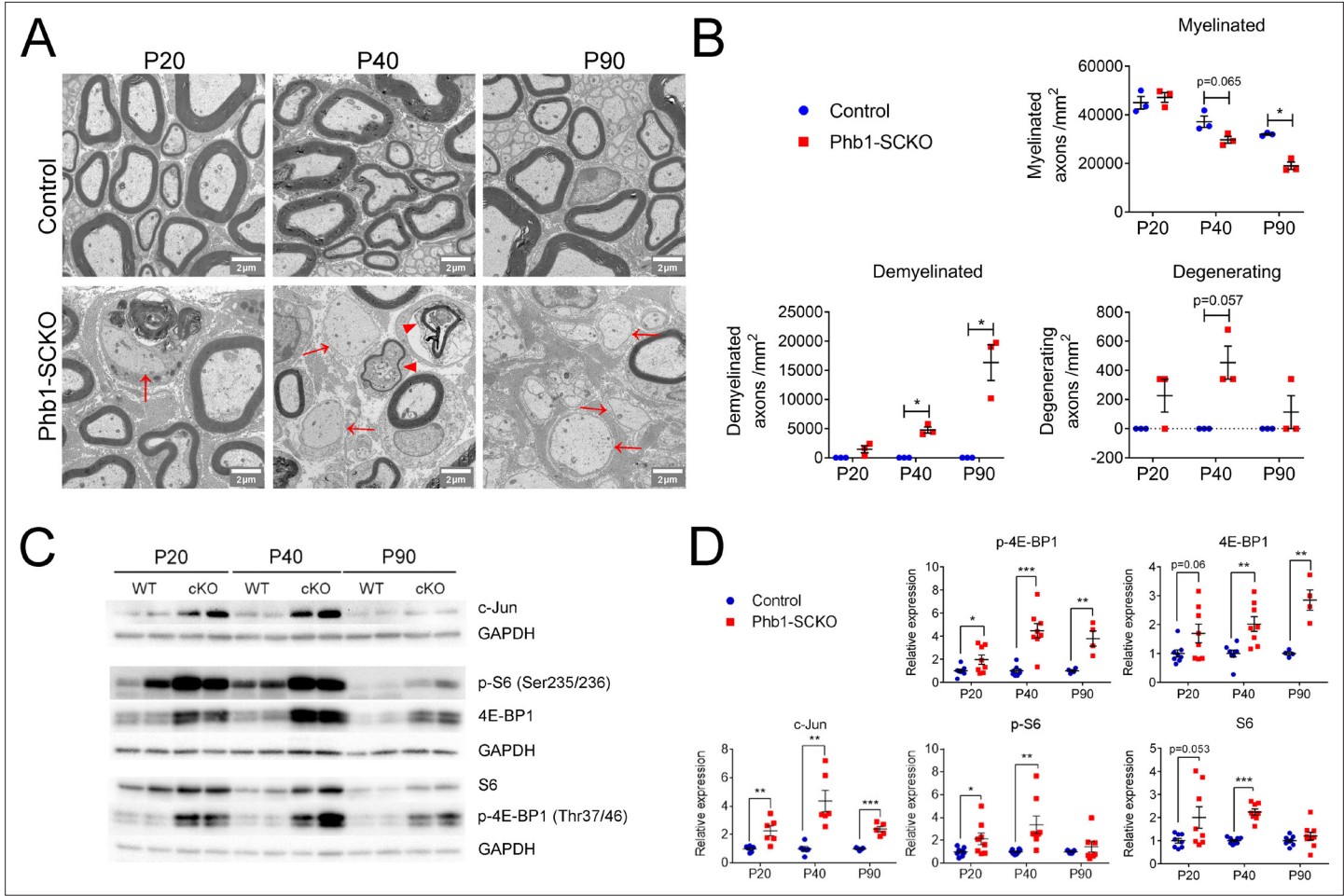

**Figure 1.** The mTORC1 and c-Jun pathways are activated upon deletion of Phb1. (**A**) Representative electron micrographs of sciatic nerves demonstrating the demyelinating phenotype of Phb1-SCKO mice. Note the presence of demyelinated axons (arrows) and degenerating axons (arrowheads). (**B**) Quantification of myelinated, demyelinated and degenerating axons at the analyzed ages. There is progressive demyelination and axonal degeneration in the sciatic nerves of Phb1-SCKO animals. N = 3 animals per genotype. Two-way ANOVA. Myelinated: F (2, 8) interaction = 7.544, p = 0.0144; F (1.941, 7.763) time = 55.74, p < 0.0001; F (1.941, 7.763) group = 21.91, p = 0.0094. Demyelinated: F (2, 8) interaction = 14.25, p = 0.0023; F (1.011, 4.042) time = 14.25, p = 0.0191; F (1, 4) group = 113.5, p = 0.0004. Degenerating: F (2,8) interaction = 1.75, p = 0.234; F (1, 4) time = 1.75, p = 0.2564; F (1, 4) group = 49, p = 0.0022. (**C**) Representative western blot from sciatic nerve lysates reveal that deletion of *Phb1* leads to upregulation of c-Jun and increased total and phosphorylated levels of the mTORC1 targets S6 and 4E-BP1. (**D**) Quantitative analysis of the relative expression of the proteins in (**C**). N = 4–9 animals per genotype. Unpaired two-tailed t-test. p-4E-BP1 [P20 (t = 2.264, df = 14), P40 (t = 5.337, df = 14), P90 (t = 4.228, df = 6)]; 4E-BP1 [P20 (t = 2.013, df = 14), P40 (t = 3.625, df = 14), P90 (t = 5.199, df = 6)]; c-Jun [P20 (t = 3.53, df = 10), P40 (t = 4.34, df = 10), P90 (t = 7.172, df = 8)]; p-S6 [P20 (t = 2.186, df = 14), P40 (t = 3.086, df = 14), P90 (t = 0.936, df = 12)]; S6 [P20 (t = 2.104, df = 14), P40 (t = 8.838, df = 14), P90 (t = 1.084, df = 16)]; * p < 0.05, ** p < 0.01, *** p < 0.001.

The online version of this article includes the following source data and figure supplement(s) for figure 1:

**Source data 1.** Raw data and annotated uncropped western blots from *Figure 1*.

**Source data 2.** Raw data and statistical analyses for data in *Figure 1*.

**Figure supplement 1.** Akt and mTOR phosphorylation are not significantly altered by deletion of *Phb1* in SCs.

**Figure supplement 1—source data 1.** Raw data and annotated uncropped western blots from *Figure 1—figure supplement 1*.

**Figure supplement 1—source data 2.** Raw data and statistical analyses for data in *Figure 1—figure supplement 1*.

**Figure supplement 2.** c-Jun and the mTORC1 pathway continue to be elevated in P60 Phb1-SCKO mice.

**Figure supplement 2—source data 1.** Raw data and annotated uncropped western blots from *Figure 1—figure supplement 2*.

**Figure supplement 2—source data 2.** Raw data and statistical analyses for data in *Figure 1—figure supplement 2*.

**Figure supplement 3.** Ratios of phosphorylated to total 4E-BP1 and S6 are only slightly altered by deletion of *Phb1*.

**Figure supplement 3—source data 1.** Raw data and statistical analyses for data in *Figure 1—figure supplement 3*.

**Figure supplement 4.** Macrophages contribute minimally to c-Jun and p-S6 in the sciatic nerves of Phb1-SCKO mice.

EBPβ (*Horibe and Hoogenraad, 2007*). On the other hand, mTORC1 was shown to be upstream of the integrated stress response (ISR) in muscle, regulating the progression of a mitochondrial myopathy (*Khan et al., 2017*).

For this reason, we sought to investigate if c-Jun and mTORC1 were activated downstream of perturbations to SC mitochondria. We treated primary rat SCs with compounds that affect different aspects of mitochondrial function: Carbonyl cyanide-p-trifluoromethoxyphenylhydrazone (FCCP), an ionophore that dissipates the mitochondrial membrane potential; Oligomycin, an inhibitor of ATP synthase; or Antimycin A, an inhibitor of mitochondrial complex III. We then evaluated the expression levels of c-Jun and downstream targets of the mTORC1 pathway (4E-BP1 and S6), as well as components of the other pathways we previously demonstrated to be altered in Phb1-SCKO mice (*Della-Flora Nunes et al., 2021*): eIF2α (which is phosphorylated downstream of different cellular stressors to activate the ISR), BiP (a chaperone also known as HSPA that is upregulated in response to stress in the endoplasmic reticulum, ER), and Opa1 (a protein essential for mitochondrial fusion that, under situations of mitochondrial damage, is proteolytically cleaved, inhibiting mitochondrial fusion). Short-term treatment (24 h) with these compounds resulted only in minor changes in these pathways (*Figure 2—figure supplement 1A and B*). Cells treated with FCCP for 24 hr presented with elevated levels of p-eIF2α and reduced ratio between the long and short isoforms of Opa1 (*Figure 2—figure supplement 1A and B*). This is in line with previous reports showing that the ISR can quickly be activated directly downstream of compromised mitochondria (*Viader et al., 2013*; *Mick et al., 2020*). Oligomycin and Antimycin A applied to SCs for 24 hr were unable to elicit significant changes in the conditions tested. On the other hand, a 7-day treatment of SCs with Oligomycin or Antimycin A triggered robust stimulation of the ISR (*Figure 2A and B* and *Figure 2—figure supplement 2*). Furthermore, this lengthier treatment regimen also induced a potent activation of the mTORC1 pathway (*Figure 2A and B* and *Figure 2—figure supplement 2A*). Hence, it is possible that the mTORC1 pathway participates in the SC adaptation to long-term mitochondrial impairments.

Next, we interrogated whether c-Jun or mTORC1 activity were associated with presence of compromised mitochondrial network in vivo. For this experiment, we analyzed Phb1-SCKO animals in which SC mitochondria were genetically labeled with the PhAM reporter. The PhAM mouse line contains a floxed STOP construct coding for a mitochondrially targeted version of the Dendra2 fluorophore (*Pham et al., 2012*). We previously reported that, at P40, about 20 % of SCs in Phb1-SCKO mice show disrupted mitochondrial network, especially away from the cell body, where PhAM is almost completely undetectable (*Della-Flora Nunes et al., 2021*). Staining of individual myelinated fibers with c-Jun indicated that c-Jun immunoreactivity was significantly associated with the presence of compromised mitochondria (*Figure 3A and B*). Moreover, 78.81 % of all SCs with damaged mitochondria showed strong c-Jun nuclear expression, while only 23.15 % of SCs with reasonably well-preserved mitochondria stained positive for c-Jun (*Figure 3C*). In contrast, we did not identify any association between mitochondrial loss and expression of p-S6 (*Figure 3D–F*). Similar results were also found in nerves of 90-day-old Phb1-SCKO mice (*Figure 3—figure supplement 1*).

Thus, our data supports the hypothesis that both c-Jun and mTORC1 can be activated downstream of perturbations to mitochondria under certain circumstances, raising the possibility that c-Jun and mTORC1 may participate in the response to mitochondrial stress in SCs. The discrepancy in our in vitro and in vivo data may be because of the already elevated c-Jun expression in SCs in vitro (SCs in culture show an immature phenotype *Schmid et al., 2014*; *Morgan et al., 1991*). In addition, mitochondrial loss is a late event in vivo, while mTORC1 activation occurs relatively quickly, after only 1 week in vitro. Thus, mTORC1 and c-Jun may be activated in different cells or at different stages of the SC response to mitochondrial damage.

## c-Jun and mTORC1 are associated with demyelination in Phb1-SCKO mice

Given the possibility that c-Jun and mTORC1 could be activated downstream of mitochondrial dysfunction, and the importance of these pathways for formation of repair SCs, we asked if demyelination in Phb1-SCKO mice was preferentially happening in SCs with overactive c-Jun or mTORC1. With this goal, we immunostained individual myelinated fibers of Phb1-SCKO animals for myelin proteins (P0 and MBP) in conjunction with c-Jun or p-S6. We then identified SCs undergoing demyelination by the presence of myelin fragments inside SCs (myelin ovoids). In our analysis, all the evaluated fibers

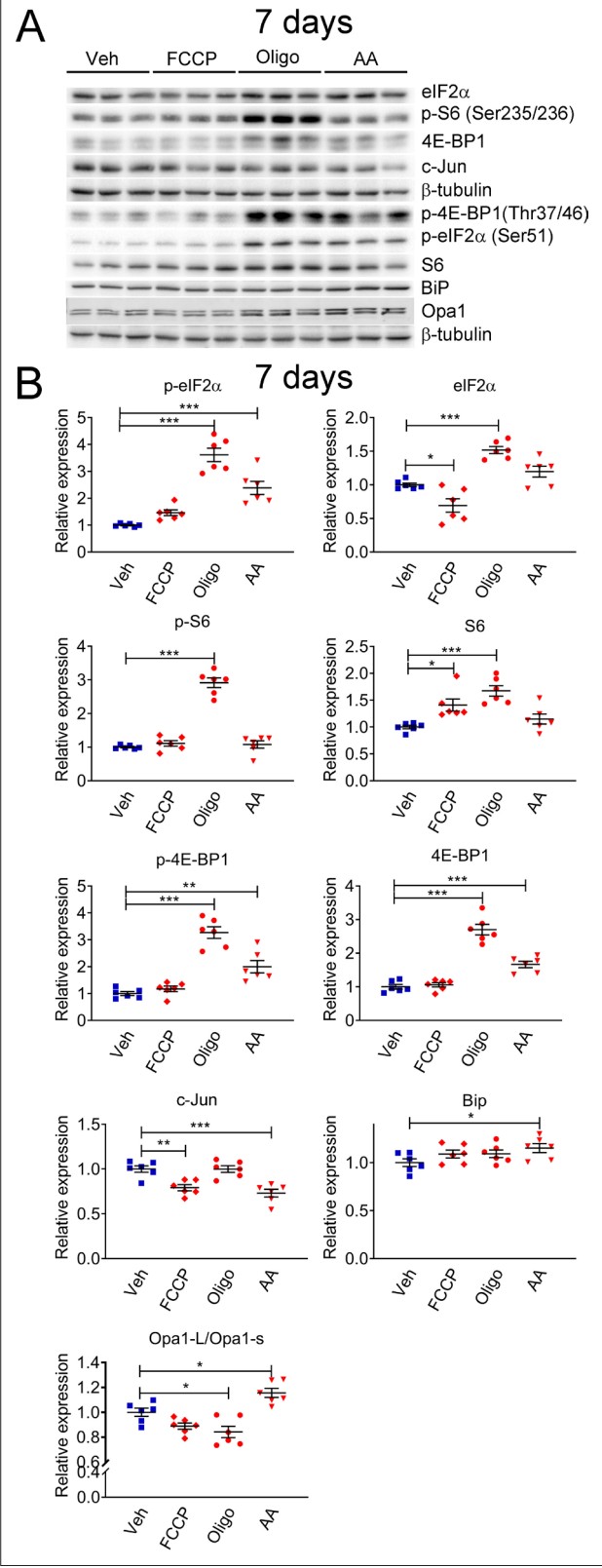

**Figure 2.** Mitochondrial perturbations in SCs in vitro lead to activation of the mTORC1 pathway. (**A**) Representative western blot from cell lysates of primary rat SCs treated with 5 µM FCCP, 2.5 µM oligomycin (Oligo), 10 µM antimycin A (AA) or vehicle (Veh) for 7 days. There is a robust activation of the ISR when SCs are treated long-term with oligomycin or antimycin A, conditions that also result in activation of the mTORC1 pathway. (**B**) Quantification

*Figure 2 continued on next page*

*Figure 2 continued*

of the experiments in (**A**). N = 6 wells per condition. One-way ANOVA corrected for multiple comparisons with the Dunnett method. $F_{(3, 20)}$ p-eIF2$\alpha$ = 39.18, $p < 0.0001$; $F_{(3, 20)}$ eIF2$\alpha$ = 23.9, $p < 0.0001$; $F_{(3, 20)}$ p-S6 = 87.39, $p < 0.0001$; $F_{(3, 20)}$ S6 = 11.13, $p < 0.0001$; $F_{(3, 20)}$ p-4E-BP1 = 37.53, $p < 0.0001$; $F_{(3, 20)}$ 4E-BP1 = 59.55, $p < 0.0001$; $F_{(3, 20)}$ c-Jun = 13.82, $p < 0.0001$; $F_{(3, 20)}$ Bip = 2.211, $p = 0.118$; $F_{(3, 20)}$ Opa1-L/Opa1-s = 15.09, $p < 0.0001$. * $p < 0.05$, ** $p < 0.01$, *** $p < 0.001$.

The online version of this article includes the following figure supplement(s) for figure 2:

**Figure supplement 1.** Short-term mitochondrial perturbation in SCs in vitro lead to activation of the ISR.

**Figure supplement 2.** Ratios of phosphorylated to total eIf2$\alpha$, 4E-BP1 and S6 are altered in SCs by a 7 -day treatment with compounds that affect mitochondrial function.

**Source data 1.** Raw data and annotated uncropped western blots from *Figure 2*.

**Source data 2.** Raw data and statistical analyses for data in *Figure 2*.

**Figure supplement 1—source data 1.** Raw data and annotated uncropped western blots from *Figure 2—figure supplement 1*.

**Figure supplement 1—source data 2.** Raw data and statistical analyses for data in *Figure 2—figure supplement 1*.

**Figure supplement 2—source data 1.** Raw data and statistical analyses for data in *Figure 2—figure supplement 2*.

---

containing myelin ovoids showed intense nuclear c-Jun staining (*Figure 4A–C*), suggesting a strong association between c-Jun expression and demyelination. Similarly, we also identified a correlation between p-S6 and the presence of myelin ovoids (*Figure 4D and E*), and a greater proportion of the SCs containing myelin ovoids were also positive for p-S6, although this difference did not reach statistical significance (*Figure 4F*).

We then tested if Phb1-SCKO mice showed activation of other pathways known to participate in breakdown and degradation of myelin. First, we probed Phb1-SCKO animals for genes modulated by c-Jun after nerve injury. After nerve damage, c-Jun leads to upregulation of genes involved in regeneration and trophic support, such as glial-derived neurotrophic factor (*Gdnf*) and Sonic hedgehog (*Shh*), and to increased expression of the transcription factor (*Olig1*), all of which are specific to the repair SC and are not expressed during SC development (*Arthur-Farraj et al., 2012*). On the other hand, as myelin gets degraded, and due to the antagonistic actions of c-Jun on EGR2 (*Parkinson et al., 2008*), there is downregulation of several molecules. One is *Cdh1*, coding for Cadherin-1, also called E-cadherin (*Arthur-Farraj et al., 2012*), an adhesion molecule present in adherens junctions that stabilizes the non-compact regions of myelin (*Young et al., 2002*; *Tricaud et al., 2005*). Also downregulated are the myelin genes Myelin basic protein (*Mbp*) and Myelin protein zero (*Mpz*). We found that the mRNA expression of the majority of the genes evaluated is altered in Phb1-SCKO mice in the expected direction: upregulation of *Gdnf* and *Shh*; downregulation of *Cdh1, Mbp* and *Mpz* (*Figure 4—figure supplement 1*). We did not, however, identify any change in expression of *Olig1*. Among these genes, *Gdnf* is a direct c-Jun target (*Fontana et al., 2012*), *Shh* has a c-Jun binding site on its enhancer (*Hung et al., 2015*), and the *Olig1* enhancer has a binding site for Runx2 (a transcription factor proposed to mediate activation of some injury-responsive genes downstream of c-Jun *Hung et al., 2015*). Finally, *Mpz* and *Mbp* are directly regulated by EGR2 (*LeBlanc et al., 2006*; *Denarier et al., 2005*), which is known to have a cross-antagonistic relationship with c-Jun (*Parkinson et al., 2008*). Although this provides further evidence for involvement of c-Jun in the nerve pathology of Phb1-SCKO mice, we cannot rule out that other pathways are also regulating the expression of the evaluated genes.

Phb1-SCKO mice also showed overexpression of Mixed lineage kinase domain-like (MLKL), a protein recently implicated in dismantling the myelin sheath after nerve injury (*Ying et al., 2018*; *Figure 4—figure supplement 1B and D*). Moreover, deletion of *Phb1* in SCs also resulted in upregulation of the autophagic machinery, which is important for myelinophagy and is regulated by c-Jun after nerve injury (*Gomez-Sanchez et al., 2015*; *Figure 4—figure supplement 1C and D*).

Taken together, these results indicate that nerves of Phb1-SCKO mice show molecular similarities to nerves undergoing myelin breakdown after nerve injury, including upregulation of myelin degradative pathways such as MLKL and autophagy, increased expression of the *Gdnf* and *Shh* trophic factors, and reduced levels of the *Mbp* and *Mpz* myelin genes. After nerve injury, many of these

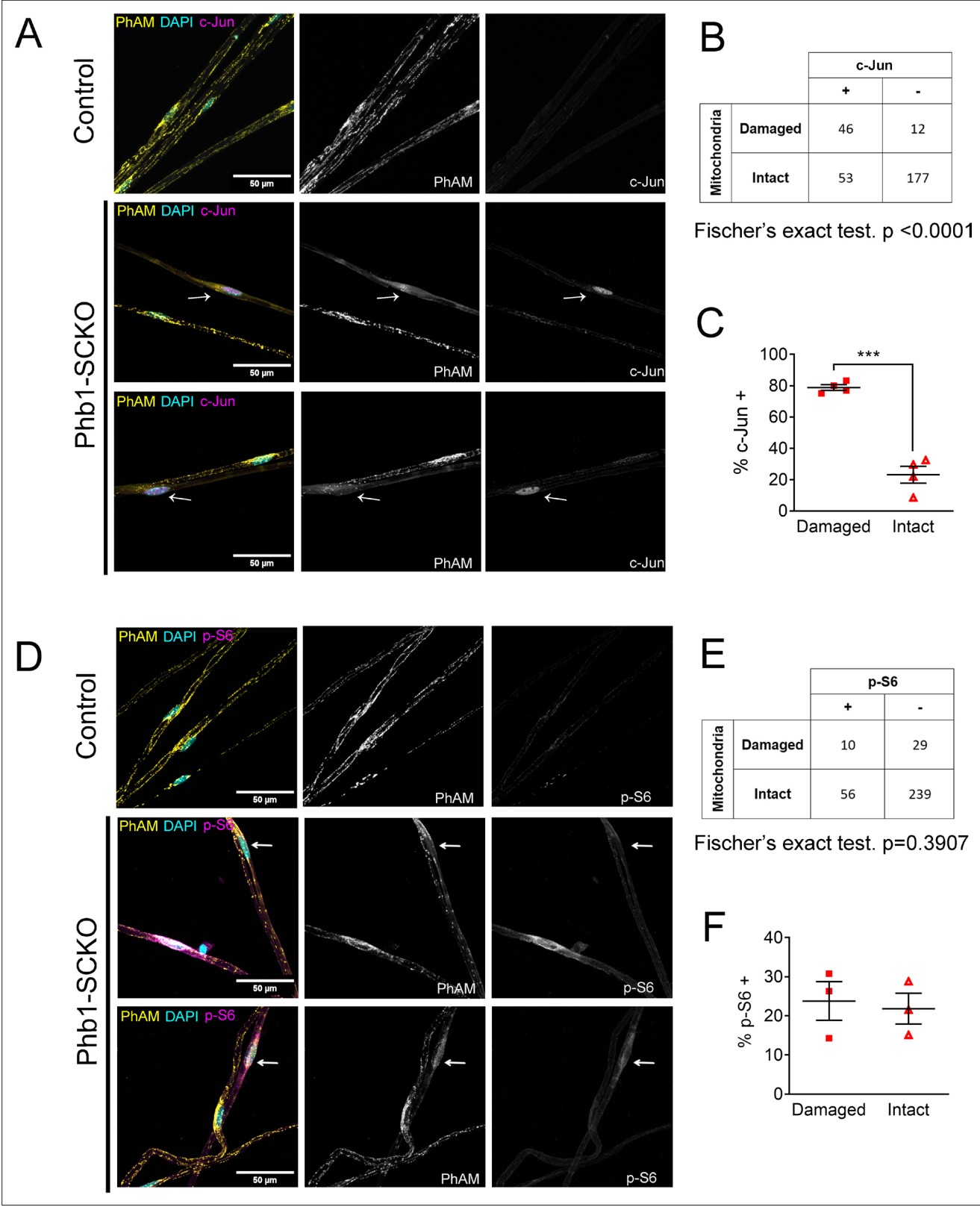

**Figure 3.** c-Jun expression is associated with mitochondrial loss in vivo, but mTORC1 activation is not at P40. (**A**) Immunofluorescence for c-Jun in teased fibers from sciatic nerves of P40 animals expressing PhAM in SC mitochondria. Note that SCs of Phb1-SCKO mice that show perturbation to their mitochondrial network (arrows) tend to also show high nuclear c-Jun expression (magenta). (**B**) There is an association between mitochondrial damage and c-Jun staining in SCs of Phb1-SCKO mice. N = 4 animals. Fischer's exact test. (**C**) The majority of SCs of Phb1-SCKO mice that lack PhAM

*Figure 3 continued on next page*

*Figure 3 continued*

expression away from the cell body show positive staining for c-Jun. N = 4 animals. Paired two-tailed t-test (t = 14.09, df = 3) (**D**) Immunofluorescence for phosphorylated S6 ribosomal protein (**p–S6**) in teased fibers from sciatic nerves of animals expressing PhAM. Arrows show two SCs with damaged mitochondria, the one on top was considered p-S6 -, while the one on the bottom was classified as p-S6 +. (**E**) There is no correlation between mitochondrial damage and p-S6 staining in SCs of Phb1-SCKO mice. N = 3 animals. Fischer's exact test. (**F**) The percentage of SCs labeed with p-S6 is constant regardless of the status of their mitochondrial network visualized by PhAM. N = 3 animals. Paired two-tailed t-test (t = 0.236, df = 2). *** p < 0.001.

The online version of this article includes the following figure supplement(s) for figure 3:

**Source data 1.** Raw data and statistical analyses for data in *Figure 3*.

**Figure supplement 1.** At P90, c-Jun expression continues to be associated with mitochondrial loss in vivo, but mTORC1 activation is not.

**Figure supplement 1—source data 1.** Raw data and statistical analyses for data in *Figure 3—figure supplement 1*.

effects depend on the activation of the mTORC1/c-Jun axis. Moreover, we found a strong association between demyelination and activation of both mTORC1 and c-Jun in Phb1-SCKO mice. Given these results, it is tempting to hypothesize that c-Jun and mTORC1 could be key pathways orchestrating the demyelination process in Phb1-SCKO mice.

## The ISR has minor effects on the other evaluated pathways

We recently showed that the ISR is likely a beneficial response in Phb1-SCKO mice (*Della-Flora Nunes et al., 2021*). Activation of the ISR frequently leads to alterations in the mTORC1 pathway (*Ryoo and Vasudevan, 2017*; *Zhang et al., 2019*). Thus, we asked whether the ISR is upstream of the pathways analyzed in this study. Inhibition of the ISR using a daily injection of 2.5 mg/kg ISRIB as previously reported (*Della-Flora Nunes et al., 2021*) did not significantly alter p-S6, c-Jun or Opa1 expression in Phb1-SCKO mice (*Figure 5*). Nonetheless, this treatment was sufficient to result in a small reduction in levels of p-4E-BP1 and 4E-BP1 (*Figure 5B and C* and *Figure 5—figure supplement 1A*).

## c-Jun may participate in the demyelination process, but ablation of *Jun* in Phb1-SCKO mice is not sufficient to ameliorate the neuropathy phenotype

In order to test the hypothesis that c-Jun coordinates demyelination in Phb1-SCKO mice, we crossed those animals to *Jun* floxed mice. At P40, presence of one (Phb1-SCKO; JUN Het) or two *Jun* floxed alleles (Phb1; JUN SCKO) resulted in a significant and dose-dependent reduction in c-Jun protein levels in sciatic nerves (*Figure 6A*). As expected, this reduction in c-Jun also led to a dose-dependent reduction in the number of demyelinated fibers and myelinophagy events (*Figure 6B and C*). Nevertheless, deletion of *Jun* in Phb1-SCKO mice also caused a dose-dependent reduction in the number of myelinated axons in tibial nerves (*Figure 6B and C*). In addition, nerves of Phb1; JUN SCKO contained bundles with large non-myelinated axons and a visible division of the nerve in smaller fascicles (hyperfasciculation) (*Figure 6B and D*). These phenotypes suggest that ablation of *Jun* in SCs already lacking *Phb1* may exacerbate the mild developmental defects that we previously observed in Phb1-SCKO mice (*Della-Flora Nunes et al., 2021*). As a consequence, *Jun* deletion is not sufficient to rescue the motor deficits of Phb1-SCKO mice in the rotarod (*Figure 6E*).

To investigate if c-Jun was important to modulate other pathways of interest, we probed the protein levels of eIF2α, p-eIF2α, 4E-BP1, p-4E-BP1, S6, p-S6, Opa1, and BiP. Deletion of *Jun* in Phb1-SCKO mice had a dose-dependent effect on levels of p-4E-BP1 and also reduced p-S6 when both *Jun* alleles were deleted, suggesting that c-Jun may partially modulate mTORC1 (*Figure 6—figure supplement 1A and B*). A similar effect was observed on BiP levels (*Figure 6—figure supplement 1A and B*). c-Jun, however, does not seem to be important to regulate the ISR, since p-eIF2α levels are unaltered in Phb1-SCKO mice upon *Jun* deletion (*Figure 6—figure supplement 1A and B*). In agreement with this finding, ablation of *Jun* is also unable to alter the mRNA levels of ATF4 targets (*Asns*, *Chac1*, *Pck2*, and *Ddit3*, also known as *Chop*), which are upregulated downstream of p-eIF2α during ISR in Phb1-SCKO mice (*Figure 6—figure supplement 1C*).

Mice containing two floxed *Jun* alleles and one floxed *Phb1* allele (JUN SCKO; Phb1 Het) are statistically indistinguishable from control mice in all the aforementioned analyses (*Figure 6—figure*

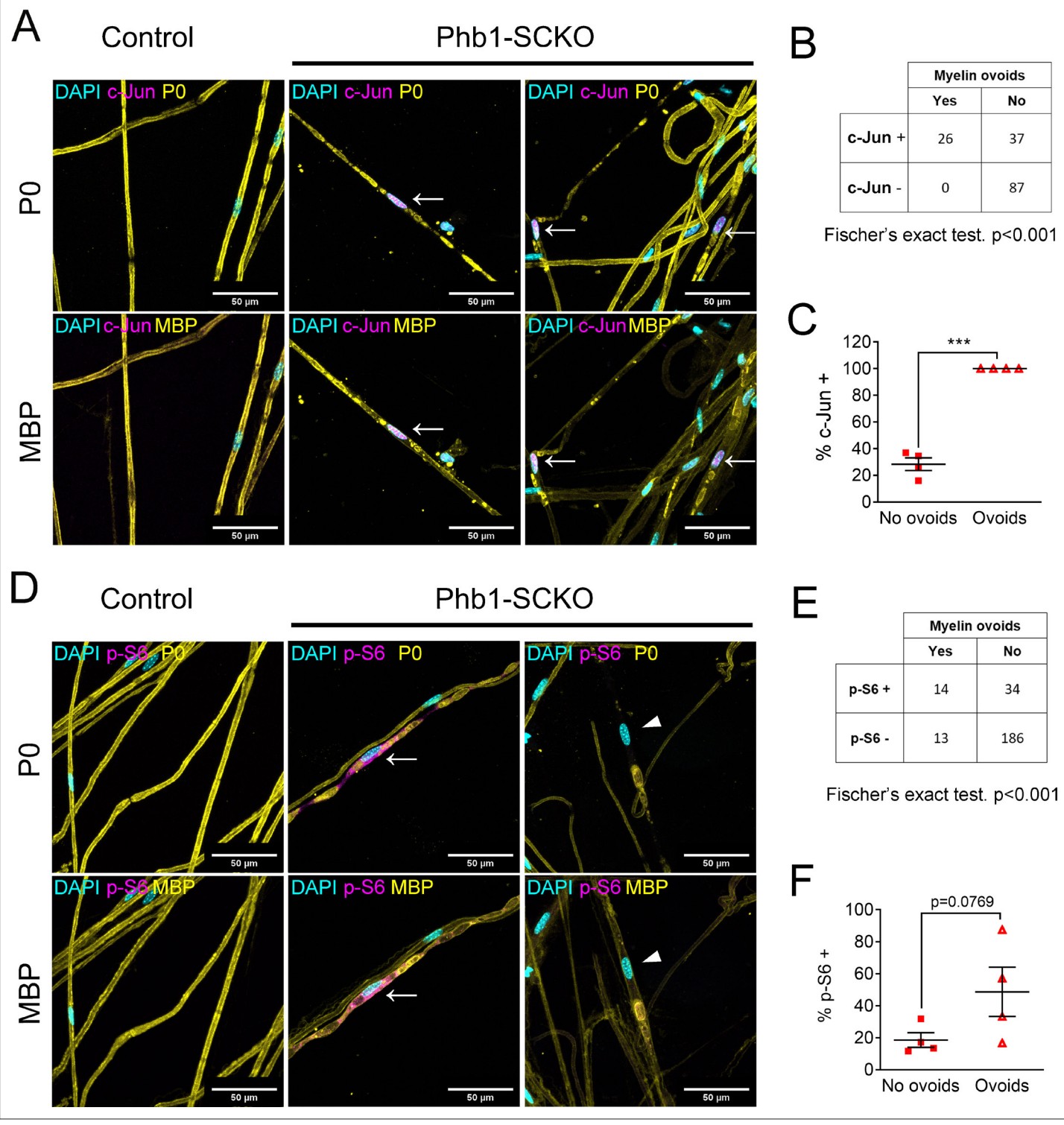

**Figure 4.** Activation of the mTORC1/c-Jun axis is associated with demyelination. (**A**) Teased fibers from tibial nerves of 40-day-old mice stained for myelin proteins (MBP and P0) and c-Jun. All cells containing myelin ovoids were also c-Jun positive (arrows). (**B**) c-Jun immunoreactivity and presence of myelin ovoids are associated. N = 4 animals. Fischer's exact test. (**C**) Cells that present with myelin ovoids show a higher percentage of c-Jun immunoreactivity. N = 4 animals. Paired two-tailed t-test (t = 15.05, df = 3). (**D**) Teased fibers from tibial nerves of 40-day-old mice stained for myelin proteins (MBP and P0) and phosphorylation of S6, a downstream target of mTORC1. Arrows and arrowheads show cells containing myelin avoids and that are p-S6 positive and p-S6 negative, respectively. (**E**) There is an association between p-S6 reactivity and presence of myelin ovoids. N = 4 animals.

*Figure 4 continued on next page*

*Figure 4 continued*

Fischer's exact test. (**F**) The percentage of cells positive for p-S6 tends to be higher among cells that present with myelin ovoids. N = 4 animals. Paired two-tailed t-test (t = 2.651, df = 3). *** p < 0.001.

The online version of this article includes the following figure supplement(s) for figure 4:

**Source data 1.** Raw data and statistical analyses for data in *Figure 4*.

**Figure supplement 1.** Phb1-SCKO mice show an overactive myelin breakdown machinery.

**Figure supplement 1—source data 1.** Raw data and annotated uncropped western blots from *Figure 4—figure supplement 1*.

**Figure supplement 1—source data 2.** Raw data and statistical analyses for data in *Figure 4—figure supplement 1*.

*supplement 2*), indicating that deletion of *Jun* alone is unable to elicit any of the changes observed in Phb1-SCKO; JUN Het or Phb1; JUN SCKO mice.

## Overactivation of mTORC1 is causal for demyelination in Phb1-SCKO mice

Next, we sought to explore the role of mTORC1 in the pathology observed in Phb1-SCKO mice. Considering the important developmental role of mTORC1 (*Beirowski et al., 2017*; *Figlia et al., 2017*; *Sherman et al., 2012*), it would be necessary to inhibit mTORC1 after myelination is completed. Therefore, we opted for a pharmacological treatment instead of a genetic approach. We administered Phb1-SCKO mice and controls with daily injections of the well-established mTORC1 inhibitor rapamycin from P20 to P40 (*Figure 7A*). Rapamycin binds to FK506-binding protein (FKBP12), which becomes an allosteric inhibitor of mTORC1 (*Li et al., 2014*; *Figure 7A*). Rapamycin treatment was efficient, and significantly reduced the levels of p-4E-BP1 and p-S6 in sciatic nerves of P40 Phb1-SCKO mice (*Figure 7B and C* and *Figure 7—figure supplement 1A*). Interestingly, suppression of mTORC1 activity resulted in almost complete rescue of nerve morphology in Phb1-SCKO animals at P40. Mutant mice treated with rapamycin had the same number of myelinated axons, demyelinated axons, degenerating axons and myelinophagy as their littermate controls (*Figure 7D and E*). We also confirmed the effectiveness of rapamycin to reduce demyelination by assessing electron micrographs (*Figure 7—figure supplement 2A and B*). Interestingly, we found a few aberrantly myelinated Remak bundles in Phb1-SCKO mice treated with rapamycin, a finding that was not present in control animals treated with the same drug (*Figure 7—figure supplement 2A*). Deletion of *Phb1* in Schwann cells did not alter myelin thickness on the myelinated axons that remained, and this was unaffected by rapamycin treatment (*Figure 7F*). Consistent with the results of our morphological analyses, rapamycin was also able to partially ameliorate the nerve conduction velocity in Phb1-SCKO mice (*Figure 7G*). Nonetheless, rapamycin could not improve the amplitude of the compound muscle action potential (CMAP) of mutant mice (*Figure 7G*) nor the motor deficits of Phb1-SCKO mice measured by the rotarod (*Figure 7H*). In summary, these results suggest that mTORC1 overactivation is key to induce demyelination in Phb1-SCKO mice, and that inhibition of the mTORC1 pathway can provide meaningful benefit to morphological parameters and nerve conduction velocity. It is, however, worth noting that, since we opted for a systemic treatment, effects of rapamycin in cells other than SCs could also be contributing to the observed outcome.

Similar to the studies with c-Jun, we also evaluated if other pathways of interest could be modulated by mTORC1. Treatment with rapamycin results in a trend toward reduction of c-Jun expression in Phb1-SCKO mice (*Figure 7—figure supplement 3A and B*). It also significantly reduces the protein levels of BiP and p-eIF2α (*Figure 7—figure supplement 3A and B*), and the mRNA level of *Ddit3* (*Figure 7—figure supplement 3C*). Therefore, mTORC1 may be a central pathway modulating c-Jun and the ISR in Phb1-SCKO mice.

## Discussion

About a third of all patients with mitochondrial genetic disorders develop peripheral neuropathies (*Pareyson et al., 2013*). Most commonly, these patients show axonal degeneration, but, when demyelination is present, the alterations in mitochondrial morphology concentrate in SCs rather than axons (*Ino and Iino, 2017*). Moreover, many recent reports demonstrate the importance of SC mitochondria in myelin homeostasis in the PNS (*Viader et al., 2011*; *Fünfschilling et al., 2012*; *Niemann et al.,*

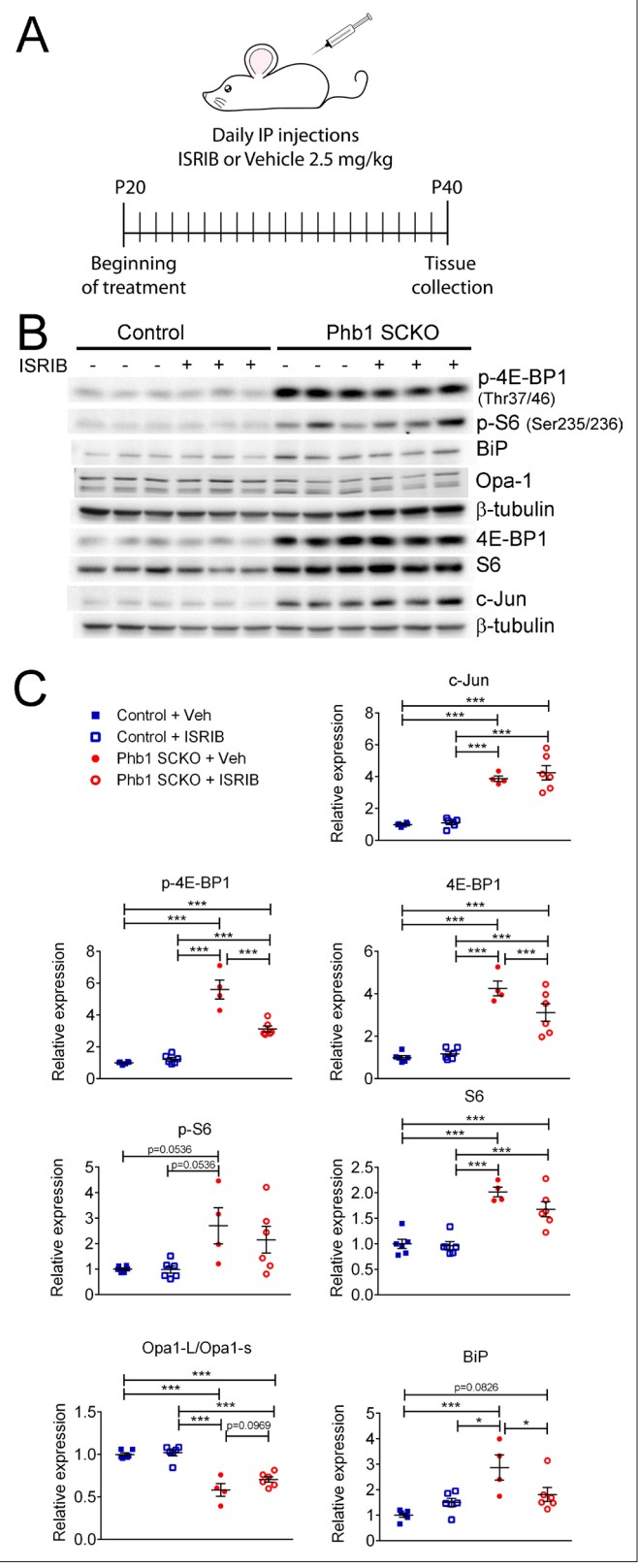

**Figure 5.** Effects of ISR on the other pathways investigated in PHB1-SCKO animals. (**A**) Schematic representation of the pharmacological treatment with the ISR inhibitor ISRIB. (**B**) Representative western blot from sciatic nerve lysates demonstrating the effect of ISRIB in P40 Phb1-SCKO mice and controls. (**C**) Quantification of the western blots illustrated in (**B**). ISRIB only seems to cause small changes in the levels of 4E-BP1 (both total and

*Figure 5 continued on next page*

*Figure 5 continued*

phosphorylated) and BiP. N = 5–6 animals per group. Two-way ANOVA corrected for multiple comparisons using the Holm-Sidak method. c-Jun: F (1, 18) interaction = 0.267, p = 0.611; F (1, 18) ISRIB = 0.823, p = 0.376; F (1, 18) group = 125.8, p < 0.0001. p-4E-BP1: F (1, 18) interaction = 31.17, p < 0.0001; F (1, 18) ISRIB = 21.56, p = 0.0002; F (1, 18) group = 180.5, p < 0.0001. 4E-BP1: F (1, 18) interaction = 5.778, p = 0.027; F (1, 18) ISRIB = 3.133, p = 0.0937; F (1, 18) group = 90, p < 0.0001. p-S6: F (1, 18) interaction = 0.459, p = 0.5066; F (1, 18) ISRIB = 0.5113, p = 0.484; F (1, 18) group = 13.23, p = 0.0019. S6: F (1, 18) interaction = 1.891, p = 0.186; F (1, 18) ISRIB = 2.861, p = 0.108; F (1, 18) group = 60.64, p < 0.0001. Opa1-L/Opa1-s: F (1, 18) interaction = 1.719, p = 0.206; F (1, 18) ISRIB = 3.334, p = 0.0845; F (1, 18) group = 85, p < 0.0001. BiP: F (1, 18) interaction = 9.354, p = 0.0068; F (1, 18) ISRIB = 1.202, p = 0.2874; F (1, 18) group = 18.28, p = 0.0005. * p < 0.05, ** p < 0.01, *** p < 0.001.

The online version of this article includes the following figure supplement(s) for figure 5:

**Figure supplement 1.** Ratios of phosphorylated to total 4E-BP1 and S6 are not affected by treatment with ISRIB.

**Source data 1.** Raw data and annotated uncropped western blots from *Figure 5*.

**Source data 2.** Raw data and statistical analyses for data in *Figure 5*.

**Figure supplement 1—source data 1.** Raw data and statistical analyses for data in *Figure 5—figure supplement 1*.

*2014*; *Wang et al., 2016*; *Della-Flora Nunes et al., 2021*). Nevertheless, the SC adaptations to mitochondrial dysfunction and the connection to demyelination remain incompletely understood.

Here, we report on a new mechanism involving mTORC1 and c-Jun that could provide a link between mitochondrial dysfunction in SCs and demyelination. Ablation of the mitochondrial protein PHB1 in SCs in mice results in continuous upregulation of c-Jun and downstream targets of mTORC1. Moreover, we demonstrate that mTORC1 can be activated in vitro by direct inhibition of mitochondrial function with oligomycin or antimycin, while c-Jun is associated with mitochondrial loss in SCs in vivo. This supports the hypothesis that c-Jun and mTORC1 are involved in the SC adaptation to mitochondrial damage. The c-Jun N-terminal Kinase (JNK) signaling pathway has been reported to modulate mitochondrial respiration and production of reactive oxygen species (ROS) (*Win et al., 2014*; *Chambers and LoGrasso, 2011*), while mTORC1 is a well-known regulator of metabolism and mitochondrial function (*Ramanathan and Schreiber, 2009*; *de la Cruz López et al., 2019*). Therefore, it is possible that mTORC1 and c-Jun also participate in the response to mitochondrial damage in other cells types. In fact, c-Jun was suggested to be involved in the activation of the mtUPR in COS-7 cells (*Horibe and Hoogenraad, 2007*), while mTORC1 was shown to participate in the response to mitochondrial dysfunction in muscle (*Khan et al., 2017*) and mouse embryonic fibroblasts (*Hardy and Pryde, 2020*).

Interestingly, we identified a consistent upregulation of 4E-BP1 and S6 (two downstream effectors of the mTORC1 pathway) during situations of mitochondrial dysfunction. To our knowledge, the consequences of overexpression of these proteins in SCs has not been explored. However, overexpression of 4E-BP1 is neuroprotective in neuronal cultures treated with brefeldin A, rotenone, maneb, or paraquat (*Dastidar et al., 2020*), compounds known to affect mitochondrial function (*Drechsel and Patel, 2008*). Importantly, these neuroprotective effects are believed to be mediated by activation of the mtUPR downstream of 4E-BP1 (*Dastidar et al., 2020*). Moreover, upregulation of 4E-BP1 is protective in pancreatic islet cells in a context of ER stress in different models of diabetes (*Yamaguchi et al., 2008*). 4E-BP1 is also overexpressed in a multitude of cancer types, inhibiting the pro-oncogene eIF4E, but also favoring tumorigenesis, especially in the context of cellular stress (*Musa et al., 2016*). Similarly, S6 is commonly upregulated in tumors, which can be important for tumor progression (*Hagner et al., 2011*; *Chen et al., 2015*). Therefore, it is possible that 4E-BP1 and S6 levels play a role in the adaptation of cells to stress.

The mechanism activating mTORC1 and c-Jun downstream of mitochondrial damage is still unknown, but we demonstrated that it is not likely to involve the Akt/PI3K pathway (because phosphorylation of Akt and of mTOR are not altered in Phb1-SCKO mice) or the ISR (since treatment of Phb1-SCKO mice with ISRIB only led to minor changes in the mTORC1 and c-Jun pathways). Even though we favor an indirect role of PHBs on the activation of the mTORC1/c-Jun axis, we cannot rule out a direct interaction. Supporting this idea, PHB2 was found to be a putative mTORC1 interactor in human T lymphoblasts (CCRF-CEM) and human embryonic kidney (HEK293) cells (*Rahman et al., 2014*), while PHB1 was found to bind to the mTOR inhibitor FK506 binding protein 8 (FKBP8) in

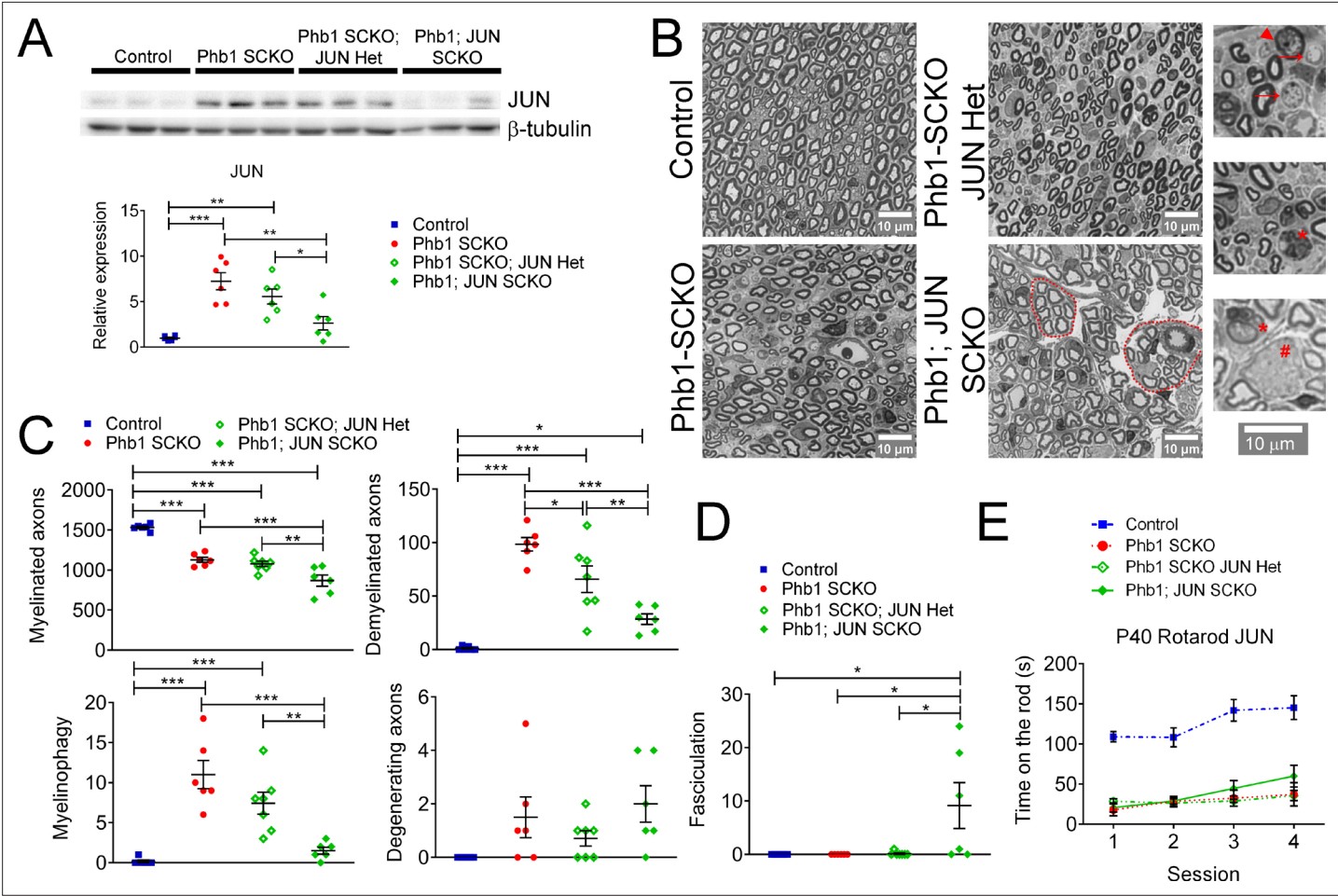

**Figure 6.** c-Jun may participate in the demyelination process in Phb1-SCKO mice, but *Jun* ablation is unable to ameliorate the behavioral phenotype. (**A**) Top: Western blot from sciatic nerve lysates illustrating the reduction in c-Jun levels when SCs of P40 Phb1-SCKO mice have one (Phb1 SCKO; JUN Het) or both JUN alleles deleted (Phb1; JUN SCKO). Bottom: Quantification of the experiment represented in the top panel. N = 6 animals per group. One-way ANOVA corrected for multiple comparisons using the Holm-Sidak method. F (3, 20) = 15.08, p < 0.0001. (**B**) Representative semithin sections from tibial nerves. Note that nerves of Phb1; JUN SCKO mice show a division of axons into smaller fascicles (dotted lines), an abnormality known as hyper-fasciculation. Nerves of these animals also frequently have large bundle structures containing axons of mixed caliber, indicative of a mild radial sorting defect (see inset). Insets: Representative images of demyelinated axons (arrows), degenerating axon (arrowhead), myelinophagy (stars) and large bundles with axons of mixed caliber (pound). (**C**) Quantification of morphological parameters from semithin images reveal that *Jun* ablation results in a reduction in demyelinated axons and myelin degradation (myelinophagy; visualized by the presence of cytoplasmic myelin debris in SCs) in Phb1-SCKO mice. Nonetheless, animals in which *Jun* has been deleted show a dose-dependent reduction in myelinated axons, suggesting that deletion of *Jun* may amplify the developmental defect observed in Phb1-SCKO animals. N = 6–7 animals per group. One-way ANOVA corrected for multiple comparisons using the Holm-Sidak method. F (3, 21) myelinated = 3.211, p = 0.0438; F (3, 21) demyelinated = 5.064, p = 0.0085; F (3, 21) myelinophagy = 2.667, p = 0.074; F (3, 21) degenerating = 2.69, p = 0.0724. (**D**) Phb1; JUN SCKO mice commonly show hyper-fasciculation. N = 6–7 animals per group. One-way ANOVA corrected for multiple comparisons using the Holm-Sidak method. F (3, 21) = 16.67, p < 0.001 (**E**) Deletion of *Jun* has no observable effect in the performance of Phb1-SCKO mice in the rotarod. N = 6 animals per group. Two-way ANOVA corrected for multiple comparisons using the Holm-Sidak method. F (9, 60) interaction = 2.654, p = 0.0117; F (3, 60) time = 19.65, p < 0.0001; F (3, 20) group = 33.63, p < 0.0001. * p < 0.05, ** p < 0.01, *** p < 0.001.

The online version of this article includes the following figure supplement(s) for figure 6:

**Source data 1.** Raw data and annotated uncropped western blots from *Figure 6*.

**Source data 2.** Raw data and statistical analyses for data in *Figure 6*.

**Figure supplement 1.** Effect of c-Jun in the other pathways altered in PHB1-SCKO animals.

**Figure supplement 1—source data 1.** Raw data and annotated uncropped western blots from *Figure 6—figure supplement 1*.

**Figure supplement 1—source data 2.** Raw data and statistical analyses for data in *Figure 6—figure supplement 1*.

**Figure supplement 2.** Deletion of *Jun* alone has no effect on the parameters analyzed.

*Figure 6 continued on next page*

*Figure 6 continued*

**Figure supplement 2—source data 1.** Raw data and annotated uncropped western blots from *Figure 6—figure supplement 2*.

**Figure supplement 2—source data 2.** Raw data and statistical analyses for data in *Figure 6—figure supplement 2*.

different cell lines (*Zhang et al., 2021*), to inhibit c-Jun N-terminal kinase (JNK) signaling in cancer cell lines (*Yang et al., 2019*) and to stimulate c-Jun expression in cells of the colon of a mouse model of colitis (*Kathiria et al., 2013*). It is worth noting that, although PHBs are mostly found in the mitochondria, they can be present in the cytosol and nucleus of some cells in specific conditions (*Thuaud et al., 2013*), which could allow them to directly interact with transcription factors and signaling cascades.

SCs present a remarkable plasticity that endows peripheral nerves with the capacity to recover from a variety of insults. After nerve injury, myelinating and non-myelinating SCs convert to a repair-promoting phenotype, allowing degradation of myelin and cell debris and stimulating axon survival and regrowth. Later, these SCs can also differentiate to form new myelin. This entire process is controlled by the transcriptional regulator c-Jun, whose upregulation requires activation of mTORC1 (for review, see *Jessen and Mirsky, 2019*). Given this particular biology of SCs, we hypothesized that activation of mTORC1 and c-Jun in the context of mitochondrial damage could inadvertently induce demyelination in Phb1-SCKO mice. In line with this hypothesis, we found a strong association between demyelination (identified by the presence of myelin ovoids) and overactivation of c-Jun or mTORC1 in these animals. Moreover, deletion of c-Jun in SCs reduced the demyelination in Phb1-SCKO mice, but also seemed to exacerbate the developmental defects observed in these animals. On the other hand, treatment of Phb1-SCKO mice with the mTORC1 inhibitor rapamycin resulted in an important rescue in nerve morphology, while also providing a significant functional benefit in nerve conduction velocity. Thus, both c-Jun and mTORC1 seem to participate in the demyelination process of Phb1-SCKO mice.

Interestingly, c-Jun and mTORC1 have also been implicated in other peripheral neuropathies. mTORC1 overactivation may be involved in the focal hypermyelination observed in Charcot–Marie–Tooth disease types 4B1 and 4B2 (CMT4B1 and CMT4B2) (*Sawade et al., 2020*), while rapamycin treatment is able to ameliorate myelination defects in a mouse model of CMT1A (*Nicks et al., 2014*). On the other hand, c-Jun was found to be upregulated in nerve biopsies from patients affected by a variety of peripheral neuropathies (*Hutton et al., 2011*). Hence, c-Jun and mTORC1 may underlie key aspects of nerve pathology. Although outside the scope of the current work, one important facet of the pathogenesis of peripheral neuropathies is the impaired trophic support from SCs to axons. c-Jun was shown to be required to prevent loss of sensory axons in a mouse model of CMT1A (*Hantke et al., 2014*), while mTORC1 is crucial to trigger a metabolic shift in SCs that supports axonal integrity in the context of acute and subacute nerve injury (*Babetto et al., 2020*). Therefore, mTORC1 and c-Jun may have opposite effects in the SC functions of myelin maintenance and axonal support, and it would be premature to conceive therapeutics targeting these pathways for peripheral neuropathies. From our results, it seems that the main beneficial effect of rapamycin on Phb1-SCKO mice is in preventing their demyelination, while deficits in axonal integrity are not altered (analysis from electron micrographs) or are only minimally improved (quantifications from semithin sections) by this treatment.

It is unlikely that c-Jun and mTORC1 are the only pathways involved in demyelination in response to mitochondrial damage in SCs, but we believe they may form an important hub controlling this process together with the ISR. Our pharmacological and genetic approaches revealed that c-Jun, mTORC1, and ISR are interconnected, with mTORC1 playing a central role and possibly modulating the other two pathways (*Figure 8*). An interesting hypothesis that we would like to explore in the future is that the global control of translation is a key response in SCs downstream of mitochondrial damage. ISR and mTORC1 (in particular its 4E-BP1 arm) are two of the most important pathways regulating cellular translation rates. Activation of the ISR through phosphorylation of eIF2α inhibits global translation and simultaneously promotes the expression of stress response genes in a response coordinated by ATF4. On the other hand, mTORC1 phosphorylates 4E-BP1 relieving its inhibition of eIF4E and promoting translation (*Ryoo and Vasudevan, 2017*). Therefore, it is interesting that treatment with rapamycin (an approach that should reduce global translation levels) was able to ameliorate the demyelination phenotype of Phb1-SCKO mice, while treatment with ISRIB (an ISR inhibitor that should increase global translation levels) was detrimental for the demyelinating pathology (*Della-Flora Nunes et al., 2021*). Moreover, it is intriguing that ISRIB was specifically able to modulate the phosphorylation levels of 4E-BP1 and had little effect on S6. This hypothesis is particularly compelling since, in a model

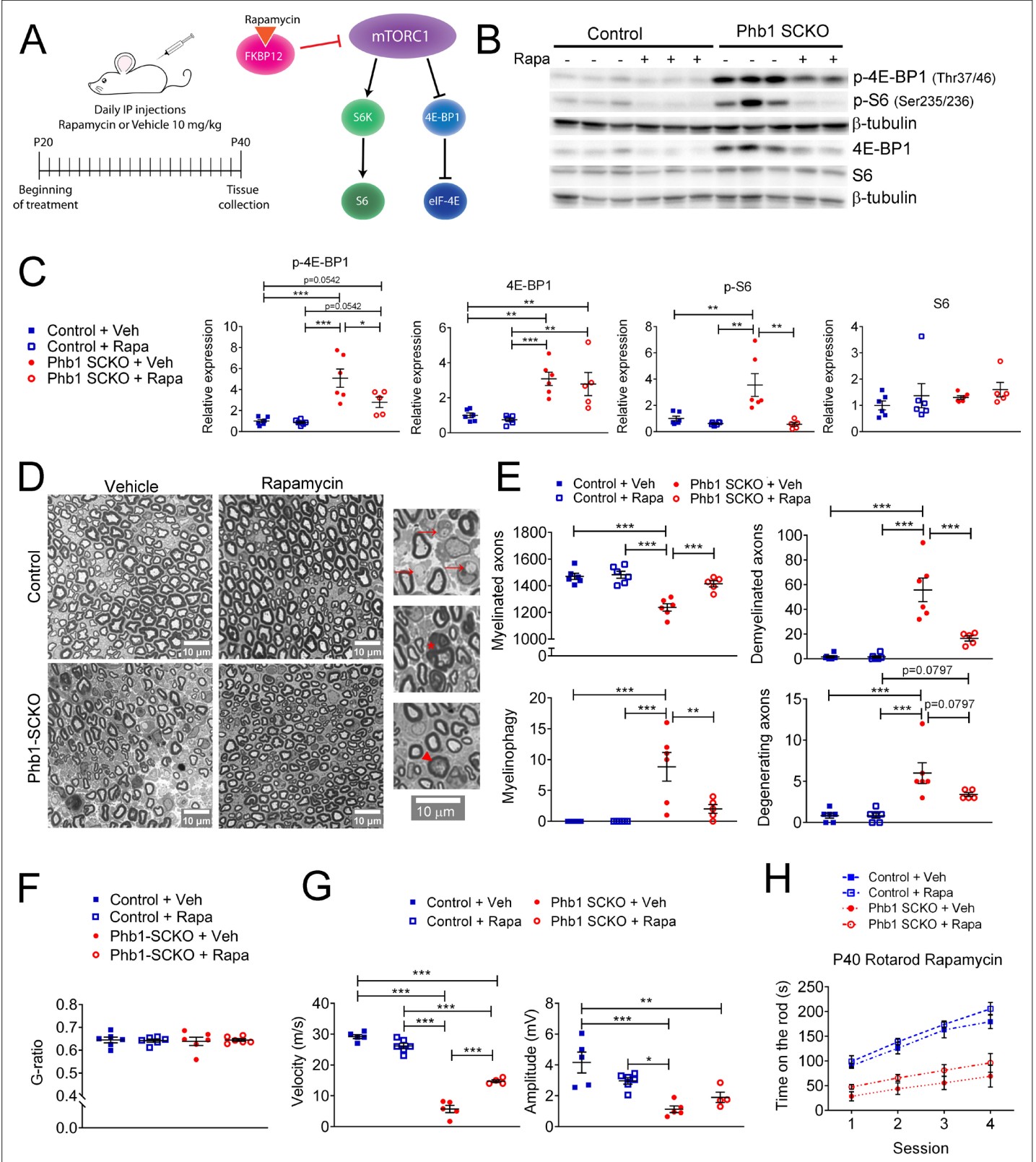

**Figure 7.** Inhibition of mTORC1 prevents demyelination in Phb1-SCKO mice. (**A**) Schematics of the rapamycin treatment (left) and mechanism of action of rapamycin (right). (**B**) Representative western blot of sciatic nerve lysates demonstrating that rapamycin treatment reduces the phosphorylation of mTORC1 targets (S6 and 4E-BP1) in Phb1-SCKO mice. (**C**) Quantification of (**B**). N = 5–6 animals per group. Two-way ANOVA corrected for multiple comparisons using the Holm-Sidak method. p-4E-BP1: F (1, 19) interaction = 4.462, p = 0.0481; F (1, 19) rapa = 5.509, p = 0.0299; F (1, 19) group =

*Figure 7 continued on next page*

*Figure 7 continued*

34.16, p < 0.0001. 4E-BP1: F (1, 19) interaction = 0.0038, p = 0.9514; F (1, 19) rapa = 0.618, p = 0.4414; F (1, 19) group = 33.58, p < 0.0001. p-S6: F (1, 19) interaction = 7.735, p = 0.0119; F (1, 19) rapa = 12.96, p = 0.0019; F (1, 19) group = 7.101, p = 0.0153. S6: F (1, 19) interaction = 0.01436, p = 0.9059; F (1, 19) rapa = 1.373, p = 0.2558; F (1, 19) group = 0.8633, p = 0.3645. (**D**) Representative tibial nerve sections of the four experimental groups. Insets: Representative images of demyelinated axons (arrows), degenerating axon (arrowhead) and myelinophagy (star). (**E**) Quantitative analysis of morphological parameters. Rapamycin treatment is able to reduce the number of demyelinated axons and SCs degrading myelin (myelinophagy), as well as increase the number of myelinated fibers in nerves of Phb1-SCKO animals. N = 5–6 animals per group. Two-way ANOVA corrected for multiple comparisons using the Holm-Sidak method. Myelinated: F (1, 19) interaction = 10.60, p = 0.0042; F (1, 19) rapa = 14.05, p = 0.0014; F (1, 19) group = 35.45, p < 0.0001. Demyelinated: F (1, 19) interaction = 14.32, p = 0.0013; F (1, 19) rapa = 14.57, p = 0.0012; F (1, 19) group = 45.02, p < 0.0001. Myelinophagy: F (1, 19) interaction = 7.333, p = 0.0139; F (1, 19) rapa = 7.333, p = 0.0139; F (1, 19) group = 18.43, p = 0.0004. Degenerating: F (1, 19) interaction = 3.344, p = 0.0832; F (1, 19) rapa = 3.344, p = 0.0832; F (1, 19) group = 29.59, p < 0.0001. (**F**) There is no alteration of myelin thickness (measured by g-ratio = axon diameter/ fiber diameter). N = 5–6 animals per group. Two-way ANOVA corrected for multiple comparisons using the Holm-Sidak method. F (1, 20) interaction = 0.1384, p = 0.7137; F (1, 20) rapa = 0.01424, p = 0.9062; F (1, 20) group = 0.01779, p = 0.8952. (**G**) Rapamycin is also able to ameliorate the nerve conduction velocity of mice lacking *Phb1*, but has no effect on CMAP amplitude. N = 4–6 animals per group. Two-way ANOVA corrected for multiple comparisons using the Holm-Sidak method. NCV: F (1, 16) interaction = 46.01, p < 0.0001; F (1, 20) rapa = 0.01424, p = 0.9062; F (1, 20) group = 0.01779, p = 0.8952. Amplitude: F (1, 16) interaction = 5.966, p = 0.0266; F (1, 16) rapa = 0.2774, p = 0.6057; F (1, 16) group = 25.98, p = 0.0001. (**H**) Phb1-SCKO mice treated with rapamycin show a trend toward improved performance in the rotarod. N = 6 animals per group. Two-way ANOVA corrected for multiple comparisons using the Holm-Sidak method. F (9, 60) interaction = 3.038, p = 0.0047; F (3, 60) time = 58.01, p < 0.0001; F (3, 20) group = 24.84, p < 0.0001. * p< 0.05, ** p < 0.01, *** p < 0.001.

The online version of this article includes the following figure supplement(s) for figure 7:

**Figure supplement 1.** Effect of rapamycin on the ratio of phosphorylated to total 4E-BP1 and S6.

**Figure supplement 2.** Rapamycin is effective at preventing demyelination in Phb1-SCKO mice, but does not seem to affect axonal degeneration.

**Source data 1.** Raw data and annotated uncropped western blots from *Figure 7*.

**Source data 2.** Raw data and statistical analyses for data in *Figure 7*.

**Figure supplement 1—source data 1.** Raw data and statistical analyses for data in *Figure 7—figure supplement 1*.

**Figure supplement 2—source data 1.** Raw data and statistical analyses for data in *Figure 7—figure supplement 2*.

**Figure supplement 3.** The role of mTORC1 in the modulation of the other pathways investigated in this work.

**Figure supplement 3—source data 1.** Raw data and annotated uncropped western blots from *Figure 7—figure supplement 3*.

**Figure supplement 3—source data 2.** Raw data and statistical analyses for data in *Figure 7—figure supplement 3*.

of CMT1B, aberrant activation of translation has already been shown to underlie demyelination in the context of ISR (*D'Antonio et al., 2013*).

In conclusion, this study reveals that mTORC1 and c-Jun may participate in the SC response to mitochondrial damage and that long-term activation of these pathways may be detrimental for myelin maintenance in the PNS. We propose this maladaptive response as a new mechanism by which perturbations in SC mitochondria trigger demyelination. This furthers our understating of how SCs respond to mitochondrial damage and could be relevant in the context of peripheral neuropathies. The link between c-Jun, mTORC1 and ISR evaluated in our study is also likely to be relevant to other cell types, and may help to advance the general understanding of cellular adaptations elicited in response to mitochondrial dysfunction.

## Materials and methods
### Animal models, genotyping, and injections

All animal procedures have been approved by the Institutional Animal Care and Use Committee (IACUC) of the Roswell Park Cancer Institute (Buffalo-NY, USA), and followed the guidelines established by the NIH's Guide for the Care and Use of Laboratory Animals and the regulations in place at the University at Buffalo (Buffalo-NY, USA). Animals were housed separated by gender in groups of at most five per cage and kept in a 12 hr light/dark cycle with water and food ad libitum. Mpz-Cre and *Phb1* floxed animals were previously described (*Feltri et al., 1999*; *He et al., 2011*). Mice were also crossed to the PhAM reporter line (Jackson Laboratories Stock No: 018385) (*Pham et al., 2012*) and to *Jun* floxed mice (*Behrens et al., 2002*). Animals were kept in a C57BL/6 and 129 mixed genetic background and analyses were performed from littermates. Animals carrying one or two floxed *Phb1* alleles but no Cre were used as controls, other than for the experiments with PhAM mice, where

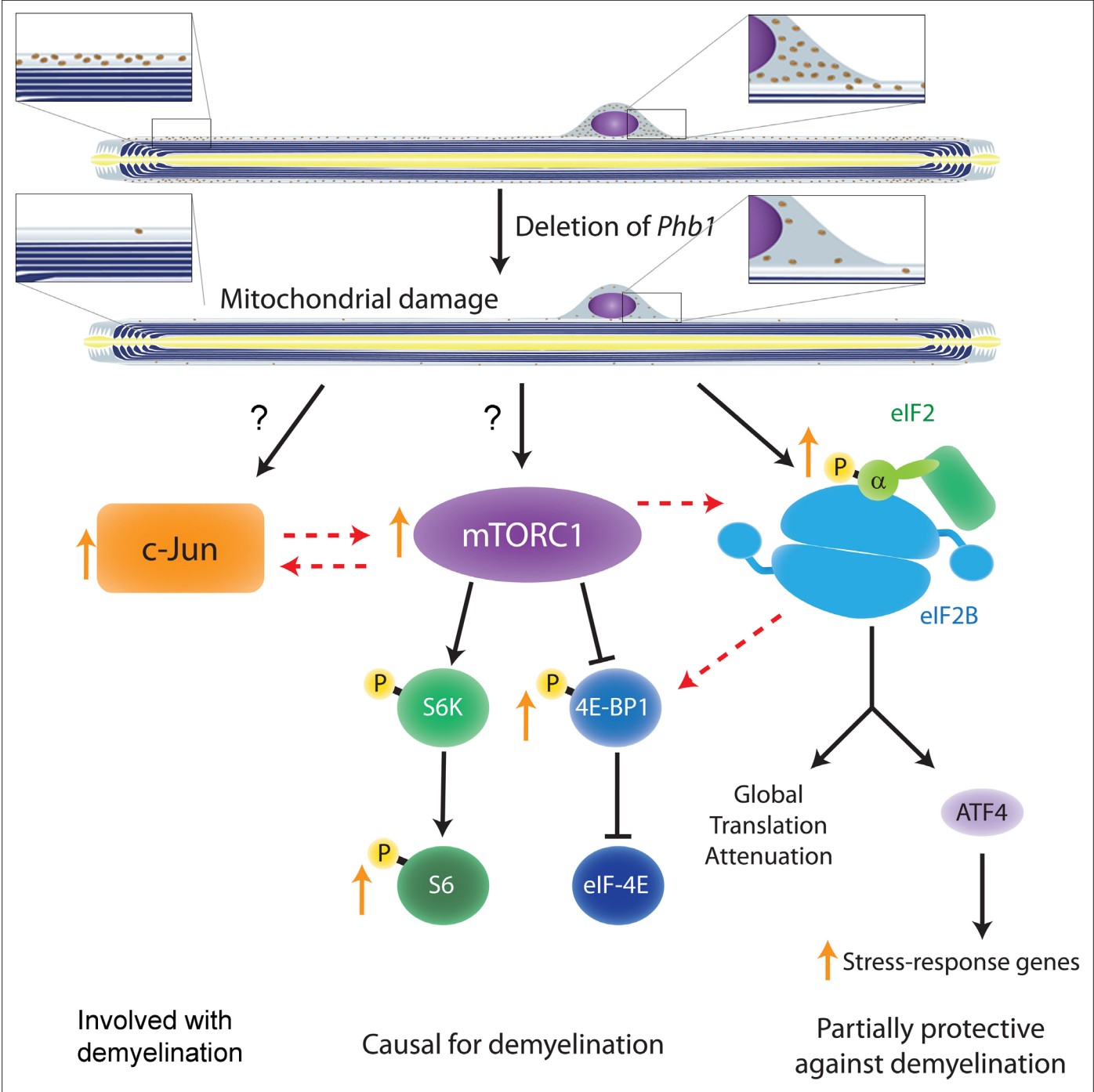

**Figure 8.** Crosstalk between the pathways investigated in the current study and in *Della-Flora Nunes et al., 2021*. Ablation of *Phb1* in SCs leads to severe perturbations to mitochondrial morphology and function, which in turn cause activation of the ISR in myelinating SCs (right). These SCs also upregulate c-Jun (left) and activate mTORC1 (center), a response that is directly or indirectly associated to the mitochondrial damage. All these pathways partially modulate each other, with mTORC1 playing the most central role and being causal for demyelination. c-Jun may also participate in demyelination and is key in the nerve repair response, while the ISR is partially protective against demyelination. Discontinuous arrows = partial effects; blunt arrows = inhibition; orange arrows = responses identified in our analyses of Phb1-SCKO mice.

Control mice were *Phb1*<sup>wt/wt</sup>; *Mpz*-Cre; PhAM, while Phb1-SCKO mice were *Phb1*<sup>fl/fl</sup>; *Mpz*-Cre; PhAM. No animals were excluded from this study. Genotyping was performed from genomic DNA as previously described for *Mpz-Cre* (*Feltri et al., 1999*), *Phb1* floxed animals (*He et al., 2011*), PhAM (*Della-Flora Nunes et al., 2021*), and *Jun* (*Parkinson et al., 2008*). Rapamycin (LC Laboratories Cat# R-5000)

was prepared as described previously (*Beirowski et al., 2017*) and administered intraperitoneally at 10 mg/kg daily from P20 to P40. ISRIB (Cayman chemicals Cat# 16258) was prepared as described previously (*Chou et al., 2017*) and administered intraperitoneally at 2.5 mg/kg daily from P20 to P40. For the above treatments, animals were randomly allocated to Vehicle or treatment group.

## Morphological assessments

Morphological analyses were performed as described previously (*Della-Flora Nunes et al., 2021*). Quantification of morphological parameters in electron micrographs in *Figure 1* used data of ten randomly selected fields at ×2900 magnification, which resulted in the evaluation of ~95–150 axons per sample. Analysis reported on *Figure 7—figure supplement 2* included about 20–30 randomly selected fields of view at ×2900 magnification, which resulted in the evaluation of ~250–450 axons per sample. Data were quantified using the cell counter plugin of ImageJ Fiji v1.52p (*Rueden et al., 2017*; *Schindelin et al., 2012*). Axons were considered to be degenerating when one of the following conditions was present: 1. visible transport defects that originated accumulation of organelles and vesicles inside the axon; 2. Axonal shrinkage resulting in blackened axon appearance under EM; and 3. Axonal swelling causing appearance of disperse distribution of microtubules and neurofilaments.

## Behavioral and electrophysiological analyses

These experiments were performed as reported before (*Della-Flora Nunes et al., 2021*).

## Cell culture

Primary rat SCs were prepared as described previously (*Brockes et al., 1979*) and were not passaged more than four times. Cells were maintained in media containing high glucose DMEM (4.5 g/L glucose) supplemented with 10 % fetal bovine serum (FBS), 2 mM L-glutamine, 100 U/mL penicillin, 100 µg/mL streptomycin, 2 ng/ml Nrg1 (human NRG1-β1 extracellular domain, R&D Systems 377-HB), and 2 µM forskolin. For the induction of mitochondrial stress, we prepared stock solutions of 10 mM FCCP (Sigma C2920) in ethanol, 40 mM Antimycin A (Sigma A8674) in ethanol and 5 mM Oligomycin (Millipore 495455) in DMSO and stored at –20 °C until use. For the experiment, 215,000 primary rat SCs were plated in each well of a 12-well dish. Two days later, media was exchanged to SC media containing 5 µM FCCP, 2.5 µM oligomycin, 10 µM antimycin A or an equivalent dose of vehicle. For the seven-day treatment, SC media was exchanged every other day with media containing a fresh dilution of the drugs. At the end of the experiment, protein was extracted and western blot was carried out as described below.

## Western blot

Western blots were performed as described previously (*Della-Flora Nunes et al., 2021*). Briefly, sciatic nerves were collected, stripped from epineurium, snap frozen in liquid nitrogen and stored at –80 °C until analysis. Nerves were pulverized and lysis was performed in buffer containing 50 mM Tris pH 7.4, 150 mM NaCl, 1 % IGEPAL CA-630, 1 mM EDTA, 1 mM EGTA, 0.1 % SDS, 0.5 % sodium deoxycholate, 1 mM sodium orthovanadate, 1 mM sodium fluoride, protease inhibitor cocktail (Sigma-Aldrich P8340), phosphatase inhibitor cocktail 2 (Sigma-Aldrich P5726) and phosphatase inhibitor cocktail 3 (Sigma-Aldrich P0044). This solution was then sonicated for 3 cycles of 20 s at 70 % power and centrifuged at 13,200× *g* for 15 min at 4 °C. Protein in the supernatant was quantified using a BCA protein assay kit and equal amounts of protein per sample were used in the SDS-PAGE. Protein was then transferred to activated PVDF membranes. After blocking with 5 % BSA in TBS solution with 0.5 % Tween (TBS-T), membranes were incubated overnight with antibodies of interest. Membranes were then rinsed in TBS-T and incubated for 1 hr with secondary antibodies. Blots were either imaged directly with Odyssey CLx infrared imaging system (Li-Cor) or developed using ECL Select (GE Healthcare) and imaged using a ChemiDoc XRS system. Quantifications were carried out in the Image lab 6.0 software (Biorad) for blots imaged with the ChemiDoc XRS or in the Image Studio Lite 5.2 (Odyssey) for blots imaged with the Odyssey CLx. The following primary antibodies were used: Opa1 1:500 (BD Biosciences Cat# 612606), β-tubulin 1:5000 (Novus Biologicals Cat# NB600-936), GAPDH 1:5000 (Sigma Cat# G9545), eIF2α 1:500 (Cell signaling Cat# 5324), p-eIF2α 1:500 (Cell signaling Cat# 3398), Bip 1:500 (Novus Biologicals Cat# NB300-520), p-4E-BP1 1:500 (Cell signaling Cat# 2855), 4E-BP1 1:500 (Cell signaling Cat# 9644), p-S6 1:500 (Cell signaling Cat# 4858), S6 1:500 (Cell signaling Cat#

2217), c-Jun 1:500 (Cell signaling Cat# 9165), p-AKT 1:500 (Cell signaling Cat# 9271); AKT 1:500 (Cell signaling Cat# 9272); MLKL 1:500 (Abgent Cat# AP14272b); LC3 1:500 (Cell signaling Cat# 12741); Atg7 1:500 (Cell signaling Cat# 8558); p62 1:500 (Enzo Life Sciences Cat# BML-PW9860); mTOR 1:1000 (Cell signaling Cat# 2983), p-mTOR 1:1000 (Cell signaling Cat# 5536). GAPDH or β-tubulin were used as loading controls to normalize protein expression. Raw files (on Image lab or Image Studio format), as well as annotated uncropped blots are presented as **Source data files** associated to each figure.

## Immunofluorescence

Conventional immunofluorescence experiments were performed as previously described (*Della-Flora Nunes et al., 2021*) using 10 -µm-thick longitudinal sections of sciatic nerve obtained using a cryostat. For co-staining of F4/80 and c-Jun at least three fields per animal were imaged at ×40 magnification using a confocal microscope Leica SP5II running the LAS AF 2.7.9723.3 software (Leica). For co-staining of F4/80 and p-S6 at least three fields per animal were imaged at ×40 magnification using a Zeiss ApoTome microscope (Zeiss Observer.Z1 AX10) running the AxioVision 4.8.2.0 software (Zeiss). For teasing, sciatic or tibial nerves were dissected, fixed in 4 % PFA for 30 min, washed with PBS and teased in slides coated with (3-Aminopropyl)triethoxysilane (TESPA; Sigma). Coating with TESPA was achieved by subsequently submerging glass slides in acetone for 1 min, 4 % TESPA in acetone for 2 min and two times in acetone for 30 s each. The teasing procedure consisted of placing a small portion of the nerve in a PBS droplet over the TESPA-coated slide, followed by careful mechanical separation of individual fibers, first using insulin syringes (0.3 ml 31 G x 8 mm) and then using modified insulin syringes containing insect pins (Fine science tools Cat# 26002–10) attached to their needle. The immunofluorescence procedure consisted in the permeabilization of the teased fibers for 2 min using acetone, blocking for 1 hr at room temperature, incubation with primary antibodies overnight at 4 °C, incubation with secondary antibodies for 1 hr at room temperature, counterstaining with DAPI, and mounting of slides with Vectashield (Vector Laboratories). Blocking buffer consisted of 5 % fish skin gelatin and 0.1 % Triton X-100 in 1× PBS. The following primary antibodies were used: rabbit anti-c-Jun 1:200 (Cell signaling Cat# 9165), rabbit anti-p-S6 1:200 (Cell signaling Cat# 4858), chicken anti-P0 1:300 (Aves Cat# PZO), mouse anti-MBP 1:300 (Millipore Cat# MAB384) and rat anti-F4/80 1: (Biorad Cat# MCA497GA). For staining with MBP, blocking of endogenous immunoglobulins was achieved by incubation with a 1:10 dilution of unconjugated Fab Goat anti-mouse IgG (Jackson ImmunoResearch Laboratories Cat# 115-007-003) during the blocking step. Images from teased fibers were acquired at ×40 magnification and 1.5 X zoom using a confocal microscope Leica SP5II running the LAS AF 2.7.9723.3 software (Leica). Quantifications were performed using ImageJ Fiji v1.52p (*Rueden et al., 2017*; *Schindelin et al., 2012*). Four to five fields per animal were analyzed. Classification of the mitochondrial network morphology was made blind to the experimental condition and followed a qualitative assessment: the mitochondrial network was labeled as 'damaged' when there was a clear reduction in mitochondrial density along the SC length and, specially, in regions distant from the cell body and closer to the nodes of Ranvier; in all other cases, the mitochondrial network was classified as intact.

## RNA extraction and qRT-PCR analyses

RNA was isolated and reverse-transcribed as published (*Poitelon et al., 2016*). qRT-PCR was performed as reported previously (*Della-Flora Nunes et al., 2021*) and used the Universal Probe Library (UPL, Roche) for the genes involved in the ISR and SYBR green for the genes activated by c-Jun. Primers used in this study are reported in the key resources table.

## Statistical analyses

Experiments were not randomized, but data collection and analyses were performed blind to the conditions of the experiments and genotype of the mice. However, due to the severity of the phenotype, in some analyses it was not possible to completely prevent investigators from identifying if the animal was WT or mutant. No data were excluded from the analyses. For treatments with cells or animals, allocation to groups was made randomly. No power analysis was performed, but our sample sizes are similar to those generally used in the field. The statistical test used in each analysis is reported in the legend of each figure. Data are presented as mean ± s.e.m. p-values < 0.05 were considered

to represent a significant difference, while $0.05 < p < 0.1$ was considered to represent a trend. Data were analyzed using GraphPad Prism 6.01. Raw data and output of statistical analyses are available as **source data files** associated to each figure.

## Acknowledgements

We thank Dr. Erwin F Wagner (Medical University of Vienna) for the *Jun* floxed mice. This work was funded by grant NIH-NINDS-R01NS100464 (to MLF). Generation of the *Phb1* floxed animals was originally supported by National Institutes of Health Grants HD08818 and HD07857 to BWO. YP was funded by grant NIH-NINDS-R01NS110627

## Additional information

### Funding

| Funder | Grant reference number | Author |
|---|---|---|
| National Institute of Neurological Disorders and Stroke | R01NS100464 | M Laura Feltri |
| National Institutes of Health | HD08818 | Bert W O'Malley |
| National Institutes of Health | HD07857 | Bert W O'Malley |
| National Institute of Neurological Disorders and Stroke | R01NS110627 | Yannick Poitelon |

The funders had no role in study design, data collection and interpretation, or the decision to submit the work for publication.

### Author contributions

Gustavo Della-Flora Nunes, Conceptualization, Data curation, Formal analysis, Investigation, Methodology, Validation, Visualization, Writing – original draft; Emma R Wilson, Conceptualization, Formal analysis, Investigation, Methodology, Writing – review and editing; Edward Hurley, Investigation, Methodology; Bin He, Methodology, Resources; Bert W O'Malley, Funding acquisition, Resources; Yannick Poitelon, Conceptualization, Data curation, Investigation, Methodology, Writing – review and editing; Lawrence Wrabetz, Conceptualization, Funding acquisition, Supervision, Writing – review and editing; M Laura Feltri, Conceptualization, Funding acquisition, Supervision, Validation, Writing – review and editing

### Author ORCIDs

Gustavo Della-Flora Nunes  http://orcid.org/0000-0001-9323-3556
Emma R Wilson  http://orcid.org/0000-0002-8069-0173
Edward Hurley  http://orcid.org/0000-0002-1967-8933
Yannick Poitelon  http://orcid.org/0000-0001-9868-1569
M Laura Feltri  http://orcid.org/0000-0002-2276-9182

### Ethics

All animal procedures have been approved by the Institutional Animal Care and Use Committee (IACUC) of the Roswell Park Cancer Institute (Buffalo-NY, USA), and followed the guidelines stablished by the NIH's Guide for the Care and Use of Laboratory Animals and the regulations in place at the University at Buffalo (Buffalo-NY, USA).

### Decision letter and Author response

Decision letter https://doi.org/10.7554/eLife.66278.sa1
Author response https://doi.org/10.7554/eLife.66278.sa2

## Additional files

### Supplementary files
• Transparent reporting form

### Data availability
All data generated or analysed during this study are included in the manuscript and supporting files.

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

# Appendix 1

**Appendix 1**-Key Resources Table

| Reagent type (species) or resource | Designation | Source or reference | Identifiers | Additional information |
|---|---|---|---|---|
| Genetic reagent (*Mus musculus*) | *Phb1*flox | *He et al., 2011* | | |
| Genetic reagent (*Mus musculus*) | *Mpz*-Cre | *Feltri et al., 1999* | The Jackson Laboratory - Stock No: 017927, RRID:IMSR_JAX:017927 | |
| Genetic reagent (*Mus musculus*) | PhAM | *Pham et al., 2012* | The Jackson Laboratory - Stock No: 018385, RRID:IMSR_JAX:018385 | |
| Genetic reagent (*Mus musculus*) | *Jun*flox | *Behrens et al., 2002* | | |
| Cell line (*rattus norvegicus*) | Primary Schwann cell culture | *Brockes et al., 1979* | | Primary rat Schwann cells isolated using the method of Brockes et al. Sciatic nerve samples of neonatal rats of both sexes were combined. |
| Antibody | Anti-Opa1 (Mouse monoclonal) | BD Biosciences | Cat# 612606, RRID:AB_399888 | WB (1:500) |
| Antibody | Anti-β-tubulin (Rabbit polyclonal) | Novus Biologicals | Cat# NB600-936, RRID:AB_10000656 | WB (1:5000) |
| Antibody | Anti-GAPDH (Rabbit polyclonal) | Sigma | Cat# G9545, RRID:AB_796208 | WB (1:5000) |
| Antibody | Anti-eIF2α (Rabbit polyclonal) | Cell signaling | Cat# 5324, RRID:AB_10692650 | WB (1:500) |
| Antibody | Anti-p-eIF2α (Rabbit polyclonal) | Cell signaling | Cat# 3398, RRID:AB_2096481 | WB (1:500) |
| Antibody | Anti-Bip (Rabbit polyclonal) | Novus Biologicals | Cat# NB300-520, RRID:AB_10000968 | WB (1:500) |
| Antibody | Anti-p-4E-BP1 (Rabbit polyclonal) | Cell signaling | Cat# 2855, RRID:AB_560835 | WB (1:500) |
| Antibody | Anti-4E-BP1 (Rabbit polyclonal) | Cell signaling | Cat# 9644, RRID:AB_2097841 | WB (1:500) |
| Antibody | Anti-p-S6 (Rabbit polyclonal) | Cell signaling | Cat# 4858, RRID:AB_916156 | WB (1:500) IF (1:200) |
| Antibody | Anti-S6 (Rabbit polyclonal) | Cell signaling | Cat# 2217, RRID:AB_331355 | WB (1:500) |
| Antibody | Anti-c-Jun (Rabbit polyclonal) | Cell signaling | Cat# 9165, RRID:AB_2130165 | WB (1:500) IF (1:200) |
| Antibody | Anti-p-AKT (Rabbit polyclonal) | Cell signaling | Cat# 9271, RRID:AB_329825 | WB (1:500) |
| Antibody | Anti-AKT (Rabbit polyclonal) | Cell signaling | Cat# 9272, RRID:AB_329827 | WB (1:500) |
| Antibody | Anti-MLKL (Rabbit polyclonal) | Abgent | Cat# AP14272b, RRID:AB_11134649 | WB (1:500) |
| Antibody | Anti-LC3 (Rabbit polyclonal) | Cell signaling | Cat# 12741, RRID:AB_2617131 | WB (1:500) |
| Antibody | Anti-Atg7 (Rabbit polyclonal) | Cell signaling | Cat# 8558, RRID:AB_10831194 | WB (1:500) |
| Antibody | Anti-p62 (Rabbit polyclonal) | Enzo Life Sciences | Cat# BML-PW9860, RRID:AB_2196009 | WB (1:500) |
| Antibody | Anti-mTOR (Rabbit polyclonal) | Cell signaling | Cat# 2983, RRID:AB_2105622 | WB (1:1000) |
| Antibody | Anti-p-mTOR (Rabbit polyclonal) | Cell signaling | Cat# 5536, RRID:AB_10691552 | WB (1:500) |
| Antibody | Anti-P0 (Chicken polyclonal) | Aves | Cat# PZO, RRID:AB_2313561 | IF (1:300) |
| Antibody | Anti-MBP (Mouse monoclonal) | Millipore | Cat# MAB384, RRID:AB_240837 | IF (1:300) |

*Continued on next page*

*Continued*

**Appendix 1**-Key Resources Table

| Antibody | Anti-F4/80 (Rat Polyclonal) | Biorad | Cat# MCA497GA, RRID:AB_323806 | IF (1:300) |
|---|---|---|---|---|
| Sequence-based reagent | Asns_F | *Della-Flora Nunes et al., 2021* | PCR primers | GGCCACACTGTCGTCAATC. Use UPL probe # 22 |
| Sequence-based reagent | Asns_R | *Della-Flora Nunes et al., 2021* | PCR primers | AGGAAGGAAGGGCTCCACT. Use UPL probe # 22 |
| Sequence-based reagent | Chac1_F | *Della-Flora Nunes et al., 2021* | PCR primers | GTATCACCTGCCCATGTTCC. Use UPL probe # 56 |
| Sequence-based reagent | Chac1_R | *Della-Flora Nunes et al., 2021* | PCR primers | AAGAGCTACTTCGCCTCCTTC. Use UPL probe # 56 |
| Sequence-based reagent | Pck2_F | *Della-Flora Nunes et al., 2021* | PCR primers | GGCAGAGCACATGCTGATT. Use UPL probe # 9 |
| Sequence-based reagent | Pck2_R | *Della-Flora Nunes et al., 2021* | PCR primers | GCCACGTAGCGCTTTTTC. Use UPL probe # 9 |
| Sequence-based reagent | Ddit3_F | *Della-Flora Nunes et al., 2021* | PCR primers | ACCACCACACCTGAAAGCA. Use UPL probe # 11 |
| Sequence-based reagent | Ddit3_R | *Della-Flora Nunes et al., 2021* | PCR primers | GACCTCCTGCAGATCCTCAT. Use UPL probe # 11 |
| Sequence-based reagent | Gdnf_F | *Arthur-Farraj et al., 2012* | PCR primers | CCAGTGACTCCAATATGCCTG |
| Sequence-based reagent | Gdnf_R | *Arthur-Farraj et al., 2012* | PCR primers | CTCTGCGACCTTTCCCTCTG |
| Sequence-based reagent | Shh_F | *Arthur-Farraj et al., 2012* | PCR primers | AAAGCTGACCCCTTTAGCCTA |
| Sequence-based reagent | Shh_R | *Arthur-Farraj et al., 2012* | PCR primers | TTCGGAGTTTCTTGTGATCTTCC |
| Sequence-based reagent | Olig1_F | *Arthur-Farraj et al., 2012* | PCR primers | ACCAACGTTTGAGCTTGCTT |
| Sequence-based reagent | Olig1_R | *Arthur-Farraj et al., 2012* | PCR primers | GGTTAAGGACCAGCCTGTGA |
| Sequence-based reagent | Cdh1_F | *Arthur-Farraj et al., 2012* | PCR primers | CAGGTCTCCTCATGGCTTTGC |
| Sequence-based reagent | Cdh1_R | *Arthur-Farraj et al., 2012* | PCR primers | CTTCCGAAAAGAAGGCTGTCC |
| Sequence-based reagent | Mbp_F | *Arthur-Farraj et al., 2012* | PCR primers | AATCGGCTCACAAGGGATTCA |
| Sequence-based reagent | Mbp_R | *Arthur-Farraj et al., 2012* | PCR primers | TCCTCCCAGCTTAAAGATTTTGG |
| Sequence-based reagent | Mpz_F | *Arthur-Farraj et al., 2012* | PCR primers | CGGACAGGGAAATCTATGGTGC |
| Sequence-based reagent | Mpz_R | *Arthur-Farraj et al., 2012* | PCR primers | TGGTAGCGCCAGGTAAAAGAG |
| Peptide, recombinant protein | human NRG1-β1 extracellular domain | R&D Systems | 377-HB | |
| Chemical compound, drug | FCCP | Sigma | C2920 | |
| Chemical compound, drug | Oligomycin | Millipore | 495,455 | |
| Chemical compound, drug | Antimycin A | Sigma | A8674 | |
| Chemical compound, drug | Rapamycin | LC Laboratories | R-5000 | |
| Chemical compound, drug | ISRIB | Cayman chemicals | 16,258 | |
| Software, algorithm | Prism 6.01 | GraphPad | RRID:SCR_002798 | |
| Software, algorithm | ImageJ Fiji v1.52p | *Rueden et al., 2017*; *Schindelin et al., 2012* | RRID:SCR_002285 | |

