## [Decision Letter]

**Acceptance summary:**

Using mouse models combined with experiments in cell culture, this manuscript supports and substantially extends knowledge regarding the mechanistic connections between mitochondrial damage in glia cells and demyelination in the peripheral nervous system. This paper will be of interest to investigators studying the mechanisms that control myelination, demyelination, and remyelination in health and disease.

**Decision letter after peer review:**

Thank you for submitting your article "Activation of mTORC1 and JUN by Prohibitin1 loss in Schwann cells may link mitochondrial dysfunction to demyelination" for consideration by *eLife*. Your article has been reviewed by 3 peer reviewers, and the evaluation has been overseen by a Reviewing Editor and Marianne Bronner as the Senior Editor. The following individual involved in review of your submission has agreed to reveal their identity: Julia Edgar (Reviewer #2).

Essential revisions:

1) Please provide further clarifications of their interpretations of the data and further supporting evidence for their hypothesis, as detailed in the attached reviews.

2) The authors should include a time point between P40 and P90 to further understand the time course of JUN and mTORC1 changes.

3) The significant recovery of the demyelinating phenotype and nerve conduction velocity in figure 7 were noted after blockade of the mTORC1 pathway using rapamycin in Phb1-SCKO mice. Is this attributed to an important trophic function of SC mitochondria for associated axons is disrupted in Phb1-SCKO mice? Or could rapamycin delivery at P20 be too late to rescue degenerating axons? Please clarify.

*Reviewer #1 (Recommendations for the authors):*

1. It is not clear to this reviewer why a somewhat different kind of quantitative data presentation concerning the morphological analysis is used in the current manuscript (Figure 1B) compared to the previously published and cited preprint (Della-Flora Nunes et al., 2020, Prohibitin 1 is essential to preserve mitochondria and myelin integrity in Schwann cells. Available at Research Square, Preprint (Version 1)). Why is density (X/mm2) used in the current manuscript (compare to the analysis in the previous paper)? This should be clarified in the text of the current manuscript to facilitate comparisons between the data within the two reports, along with how many axons were analyzed for each mouse / sample to determine the frequency of each feature. Furthermore, details about the morphological analysis and quantifications are required (in the Materials and methods section).

2. This reviewer suggests to include a statistical analysis between the different time points analyzed in Figure 1 to support formally the statement of a fast progressing demyelinating phenotype.

3. This reviewer suggests that the authors describe and discuss in more detail the results of the mTORC1 pathway analysis in Figure 1C/D: What is the interpretation of the significantly higher levels of total 4E-BP1 levels (P40/P60) and, in particular, of the total S6 level (P20/P40) in mutants? Why did the authors select not to determine the ratio of the p-4E-BP1/total 4E-BP1 and p-S6/S6 levels as a measure of phosphorylation/total protein? Since this ratio has not been determined, a careful wording with regard to the interpretation of the data is necessary (i.e. some of the data indicate changes in the levels of phosphorylated protein, which is not necessarily congruent with induction of changes in phosphorylation/protein unit). However, this distinction is relevant for potential limitations of the interpretations in the view of this reviewer.

4. This reviewer feels that the understanding of this rather complex manuscript would profit significantly if the authors would introduce the required features of an induction of the mTORC1 pathway thoroughly BEFORE describing the first analysis of this pathway (in the context of what is known in general (with references) and in Schwann cells / peripheral nerves specifically) and discuss all subsequent analyses of the pathway compared to this introduction. This relates in particular to the interpretation of changes in total protein levels and/or phosphorylation level changes within the different experimental settings described throughout the manuscript (see point 3).

5. Please comment on the potential contributions by other cell types than Schwann cells to the observed levels of the assayed proteins/phospho-proteins in the context of the Western blot analyses of total nerve lysates of controls and the different mutants (for example, but not exclusively, by macrophages, see Della-Flora Nunes et al., 2020, Prohibitin 1 is essential to preserve mitochondria and myelin integrity in Schwann cells. Available at Research Square, Preprint (Version 1)).

6. Figure 3. Please define arrows in (A).

7. Figure 4-Supplement 1: Please provide reference(s) that validate the "JUN targets" analyzed and discuss/indicate the specificity of these targets for the given pathway.

8. Figure 6: Please clarify the level of your conclusion since the title of sections 2.6 says "JUN may participate…" while the title of the figure legend says "JUN participates…"

9. Limitations of the chosen experimental setting need to be indicated and discussed in the manuscript with regard to the systemic use of rapamycin in vivo compared to more specific (genetic) approaches.

*Reviewer #2 (Recommendations for the authors):*

It is inferred that demyelination in this situation is not occurring secondary to SC demise. What is the evidence that the SCs remain viable over the time course of the study?

In figure 1 legend, it states that "deletion of Phb1 leads to … increased phosphorylation of the mTORC1 targets S6 and 4E-BP1". However, the relative increase in the phosphorylated protein is similar to the relative increase in total S6 and 4E-BP1 levels. If one of the aims is to demonstrate there is increased phosphorylation of these targets, it would seem appropriate to report the ratio of phosphorylated to total protein.

In terms of presentation, the blot images in Figure 1 seem not to be presented in the most logical order. Further, if the authors choose not to report the ratio of phosphorylated to total protein, the paired graphs (showing total versus phosphorylated protein) should have the same scales to improve clarity around the ratios of one to the other.

As western blotting is central to the conclusions of the work, it would seem appropriate to briefly describe the protocol, in particular in relation to the use of phosphatase inhibitors, components of the lysis buffer and antibody incubation protocols. Currently the reader is referred to a previous publication.

Possibly, replacing glucose with pyruvate would have been appropriate for the study using cultured Schwann cells and mitochondrial blockers/inhibitors. This would have prevented the cells relying (solely?) on glycolysis for ATP synthesis, and might have resulted in responses more compatible with the in vivo observations.

Western blotting of whole nerve lysate demonstrated that JUN and S6 are simultaneously upregulated/activated, yet the histology shows mitochondrial disruption/myelin ovids in JUN positive cells, but not in p-S6 positive cells. These data seem to suggest the molecules are each upregulated/activated in different fibres or at different timepoints. The data do not seem to support the suggestion that mTORC1 is activated downstream of JUN.

The final two sentences in the first paragraph of the Discussion might be more appropriately placed elsewhere, as (unless I have misunderstood their data), a protective role for the ISR is not explicitly shown in the manuscript.

*Reviewer #3 (Recommendations for the authors):*

1) Figure 3, can the authors provide data regarding the efficiency of Phb1 knockout in SCs in their mouse line? Do some Phb1-deficient SCs maintain an intact mitochondrial network?

2) Figure 4: Since these are teased nerve fibers, not adjacent sections, please describe the detailed methods for immunofluorescence detection of DAPI and protein targets (JUN, P0, MBP, p-S6).

3) Lines 332 – 335: Please provide a more detailed discussion of the differential changes in Gdnf, Shh, and Cdh1 versus Mbp and Mpz. Appropriate references should be provided here.

4) Lines 439 – 440, "deletion of Jun in Phb1-SCKO mice had a dose-dependent effect on levels of p-4E-BP1 and p-S6." While the effect on p-4E-BP1 does appear to be dose-dependent, there is not a dose-dependent effect on p-S6.

5) In the discussion, it is important to acknowledge that although Phb1 is primarily a mitochondrial protein, it has also been localized to the cytosol and nucleus under certain conditions. The nuclear localization is of considerable relevance here as it may allow Phb1 to interact with transcription factors. In addition, there appear to be reports of Phb2 interacting with *JNK* to activate c-JUN. This point may need to be discussed.

---

## [Author Response]

Essential revisions:1) Please provide further clarifications of their interpretations of the data and further supporting evidence for their hypothesis, as detailed in the attached reviews.

We have now clarified the points of concern and included new results that can respond to the reviewers’ questions.

2) The authors should include a time point between P40 and P90 to further understand the time course of JUN and mTORC1 changes.

We have now included the P60 time point. This new data is reported on Figure 1—figure supplement 2. We can conclude from this experiment that the protein expression of c-Jun and mTORC1 continues to be elevated in Phb1-SCKO mice at P60 at a similar level as P40. Therefore, c-JUN and mTORC1 are overactive before overt demyelination (P20), and continue to be elevated during the peak of demyelination (around P40 to P60) and at a late stage of the neuropathy (P90).

3) The significant recovery of the demyelinating phenotype and nerve conduction velocity in figure 7 were noted after blockade of the mTORC1 pathway using rapamycin in Phb1-SCKO mice. Is this attributed to an important trophic function of SC mitochondria for associated axons is disrupted in Phb1-SCKO mice? Or could rapamycin delivery at P20 be too late to rescue degenerating axons? Please clarify.

In Phb1-SCKO mice, clear axonal degeneration seems to happen fast; so, it is challenging to detect these events. According to our data on Figure 7, Phb1-SCKO mice treated with rapamycin from P20 to P40 showed a trend towards amelioration of axonal degeneration in tibial nerves at P40 as assessed in semithin sections. To explore this data in more detail, but also to substantiate the data on rescue of demyelination by rapamycin, we now performed an analysis of the same tissue in electron microscopy. Our new results reported in Figure 7 —figure supplement 2 suggest that, although rapamycin is efficient at reducing the demyelination in Phb1-SCKO, it did not significantly alter the axonal degeneration as quantified from the electron micrographs. Therefore, we believe that the main beneficial effect of rapamycin on nerve conduction velocity is mediated by its capacity to prevent the demyelination on Phb1-SCKO mice. However, we cannot entirely rule out the possibility that maintenance of myelin sheaths by rapamycin can also have a small indirect effect on axon survival (which could be what we picked up in our previous quantification from semithin images). We adjusted the results and Discussion sections to convey that the effect of rapamycin on axonal integrity is small or even non-present in our paradigm. We also believe that earlier inhibition of mTORC1 (before P20) could be beneficial to Phb1-SCKO mice. However, mTORC1 is known to be essential for SC proliferation during development, and, therefore, an earlier treatment could also affect myelin formation. This is one of the main reasons guiding our choice for the starting point of rapamycin application.

Reviewer #1 (Recommendations for the authors):1. It is not clear to this reviewer why a somewhat different kind of quantitative data presentation concerning the morphological analysis is used in the current manuscript (Figure 1B) compared to the previously published and cited preprint (Della-Flora Nunes et al., 2020, Prohibitin 1 is essential to preserve mitochondria and myelin integrity in Schwann cells. Available at Research Square, Preprint (Version 1)). Why is density (X/mm2) used in the current manuscript (compare to the analysis in the previous paper)? This should be clarified in the text of the current manuscript to facilitate comparisons between the data within the two reports, along with how many axons were analyzed for each mouse / sample to determine the frequency of each feature. Furthermore, details about the morphological analysis and quantifications are required (in the Materials and methods section).

Thank you for your feedback! The main reason we performed a different analysis in this manuscript is that we did not want to duplicate the data in both manuscripts since the analysis were conducted from tissue collected from the same cohort of animals. However, we did want to provide an additional phenotypic characterization of Phb1-SCKO mice in this manuscript to introduce the topic to readers without the need of referring them back to our previous manuscript. We now clarified this in the text and expanded our description of the experiment in the Materials and methods section.

2. This reviewer suggests to include a statistical analysis between the different time points analyzed in Figure 1 to support formally the statement of a fast progressing demyelinating phenotype.

Thank you for the suggestion! We modified our statistical analysis on Figure 1 to also include comparisons across time points.

3. This reviewer suggests that the authors describe and discuss in more detail the results of the mTORC1 pathway analysis in Figure 1C/D: What is the interpretation of the significantly higher levels of total 4E-BP1 levels (P40/P60) and, in particular, of the total S6 level (P20/P40) in mutants? Why did the authors select not to determine the ratio of the p-4E-BP1/total 4E-BP1 and p-S6/S6 levels as a measure of phosphorylation/total protein? Since this ratio has not been determined, a careful wording with regard to the interpretation of the data is necessary (i.e. some of the data indicate changes in the levels of phosphorylated protein, which is not necessarily congruent with induction of changes in phosphorylation/protein unit). However, this distinction is relevant for potential limitations of the interpretations in the view of this reviewer.

Thank you very much for this advice! We represented only the levels of total and phosphorylated proteins because we felt that, this way, the phosphorylated/total ratio could also be easily inferred by comparing the former results. Nonetheless, we do agree that this information is important and should be readily accessible to the reader. Therefore, we have now added the results of p-4EBP1/4EBP1 and p-S6/S6 ratios in figure supplements. We are not completely sure about the meaning of the elevation of total levels of 4E-BP1 and S6. But, although the total level of these proteins is not always reported, several manipulations to SCs lead to upregulation of these proteins:

– deletion of TSC2 leads to upregulation of 4E-BP1 (Figure 3A in – Beirowski et al., 2017)

– nerve crush causes upregulation of S6 (Figure 1A in – Norrmen et al., 2018)

– gain of function of Mek1 leads to upregulation of both 4E-BP1 and S6 (Figure 5B in – Sheean et al., 2014)

– deletion of mTOR causes upregulation of 4E-BP1 (Figure 1C in – Sherman et al., 2012)

– ablation of Fbxw7 results in increased levels of 4E-BP1 mRNA (Figure 4C in – Harty et al., 2019)

To our knowledge, the consequence of upregulation of these proteins in SCs is unknown, but upregulation of 4E-BP1 has been hypothesized to be a mechanism of adaptation to continuous activation of the mTORC1 pathway (Beirowski et al., 2017). Overexpression of 4E-BP1 was also shown to be neuroprotective I the context of mitochondrial damage (Dastidar et al., 2020) and ER stress (Yamaguchi et al., 2008) in other cell types. Furthermore, 4E-BP1 is overexpressed in a multitude of cancer types, inhibiting the pro-oncogenic eIF4E, but also favoring tumorigenesis, especially in the context of cellular stress (Musa et al., 2016). Similarly, S6 is commonly upregulated in tumors, which can be important for tumor progression (Hagner et al., 2011, Chen et al., 2015). Therefore, it is conceivable that unphosphorylated 4E-BP1 and S6 could also play important roles in adaptation of cells to stress.

To discuss the importance of total 4E-BP1 and S6 levels, we introduced a new paragraph in our discussion.

4. This reviewer feels that the understanding of this rather complex manuscript would profit significantly if the authors would introduce the required features of an induction of the mTORC1 pathway thoroughly BEFORE describing the first analysis of this pathway (in the context of what is known in general (with references) and in Schwann cells / peripheral nerves specifically) and discuss all subsequent analyses of the pathway compared to this introduction. This relates in particular to the interpretation of changes in total protein levels and/or phosphorylation level changes within the different experimental settings described throughout the manuscript (see point 3).

Thank you for your suggestion! We have now added two new paragraphs in our introduction including the background information about the mTORC1 pathway and its role in peripheral nerves.

5. Please comment on the potential contributions by other cell types than Schwann cells to the observed levels of the assayed proteins/phospho-proteins in the context of the Western blot analyses of total nerve lysates of controls and the different mutants (for example, but not exclusively, by macrophages, see Della-Flora Nunes et al., 2020, Prohibitin 1 is essential to preserve mitochondria and myelin integrity in Schwann cells. Available at Research Square, Preprint (Version 1)).

Thank you for this question! We believe that SCs account for most of the alterations in the mTORC1 and c-Jun pathways. However, due to the limitation of the antibodies, we could not co-stain these proteins with a SC marker. Since there is significant macrophage infiltration in the nerves of Phb1-SCKO mice, we now evaluated whether c-Jun or p-S6 were highly expressed in those cells. In our new results, reported on Figure 1-suplement figure 4, we report that macrophages contribute only minimally to the expression level of these proteins. Therefore, even though we cannot completely rule out that other cells contribute to the levels of c-Jun and of the mTORC1 targets, we believe that the majority of the alterations happen in SCs.

6. Figure 3. Please define arrows in (A).

Thank you! We now added this description to the figure legend. The arrows are pointing to SCs with damaged mitochondrial network.

7. Figure 4-Supplement 1: Please provide reference(s) that validate the "JUN targets" analyzed and discuss/indicate the specificity of these targets for the given pathway.

Thank you for the suggestion! We had used the term “JUN targets” as a broad term encompassing genes that were modulated after nerve injury in a c-Jun-dependent way. We have now added the below sentence to the text, clarifying the level of evidence and specificity of these targets:

“Of these, *Gdnf* is a direct c-Jun target (Fontana et al., 2012), *Shh* has a c-Jun binding site on its enhancer (Hung et al., 2015), and *Olig1* enhancer has a binding site for Runx2 (a transcription factor proposed to mediate activation of some injury-responsive genes downstream of c-Jun (Hung et al., 2015)), while *Mpz* and *Mbp* are directly regulated by EGR2 (LeBlanc et al., 2006, Denarier et al., 2005), which is known to show a cross-antagonistic relationship with c-Jun (Parkinson et al., 2008). Although this provides further evidence for involvement of c-Jun in the nerve pathology of Phb1-SCKO mice, we cannot rule out that other pathways are also regulating the expression of the evaluated genes.”

8. Figure 6: Please clarify the level of your conclusion since the title of sections 2.6 says "JUN may participate…" while the title of the figure legend says "JUN participates…"

Thank you for catching this! We want to be conservative in our conclusion because of the additional effect of c-Jun on nerve development in Phb1-SCKO mice. We changed the text in the figure legend to “JUN may participate…”.

9. Limitations of the chosen experimental setting need to be indicated and discussed in the manuscript with regard to the systemic use of rapamycin in vivo compared to more specific (genetic) approaches.

Thank you for this comment! We recognize the limitation of a systemic rapamycin treatment and, therefore, we added the following sentence to our manuscript: “It is however worth noting that, since we opted for a systemic treatment, effects of rapamycin in cells other than SCs could also be contributing to the observed outcome”. However, we believe that tackling this question genetically would be technically very difficult. Because of the importance of mTORC1 pathway in development, this would require timely knockdown/knockout of components of the mTORC1 pathway after P20 in a SC-specific way. This could probably only be achieved through in vivo viral delivery to peripheral nerves, which is usually low specificity and low efficiency.

Reviewer #2 (Recommendations for the authors):It is inferred that demyelination in this situation is not occurring secondary to SC demise. What is the evidence that the SCs remain viable over the time course of the study?

Thank you for this question! This is something that we investigated in our recently published paper on the same model (Della-Flora Nunes et al., 2021). According to the data on Supplementary Figure 6 of the referred manuscript, changes to SC apoptosis or proliferation are small and not timely to explain the severe nerve pathology observed in Phb1-SCKO mice. In addition, the balance of SC death and proliferation is maintained, with no alteration in SC numbers. We also added a sentence to the text to make this clearer to the reader.

In figure 1 legend, it states that "deletion of Phb1 leads to … increased phosphorylation of the mTORC1 targets S6 and 4E-BP1". However, the relative increase in the phosphorylated protein is similar to the relative increase in total S6 and 4E-BP1 levels. If one of the aims is to demonstrate there is increased phosphorylation of these targets, it would seem appropriate to report the ratio of phosphorylated to total protein.

Thank you very much for this comment! Following this and the suggestions from Reviewer #1, we have now added information on the ratio of phosphorylated/total levels of these proteins to figure supplements. We also revised the main text and figure legends to match our findings regarding total and phosphorylated levels of S6 and 4E-BP1, and we introduced a new paragraph discussing a possible role of the unphosphorylated form of these proteins.

In terms of presentation, the blot images in Figure 1 seem not to be presented in the most logical order. Further, if the authors choose not to report the ratio of phosphorylated to total protein, the paired graphs (showing total versus phosphorylated protein) should have the same scales to improve clarity around the ratios of one to the other.

Thank you for those suggestions. I think we partially addressed this question with our previous comment. The presentation order in Figure 1 keeps proteins of interest grouped with their respective loading control. This is the case because the data comes from different Western blot membranes. Therefore, we do not think that the order can be significantly altered.

As western blotting is central to the conclusions of the work, it would seem appropriate to briefly describe the protocol, in particular in relation to the use of phosphatase inhibitors, components of the lysis buffer and antibody incubation protocols. Currently the reader is referred to a previous publication.

Thank you for the suggestion. We have now expanded our description of the western blotting method.

Possibly, replacing glucose with pyruvate would have been appropriate for the study using cultured Schwann cells and mitochondrial blockers/inhibitors. This would have prevented the cells relying (solely?) on glycolysis for ATP synthesis, and might have resulted in responses more compatible with the in vivo observations.

This is a great idea! We also wanted to test if we could perform these experiments in a condition with high cAMP, which is known to suppress c-Jun expression (Parkinson et al., 2008). However, Gustavo (the paper’s first author) recently graduated and started a postdoctoral position in a different laboratory. Therefore, he was unable to run these experiments. We hope that someone can follow this idea in our or other laboratories in the future.

Western blotting of whole nerve lysate demonstrated that JUN and S6 are simultaneously upregulated/activated, yet the histology shows mitochondrial disruption/myelin ovids in JUN positive cells, but not in p-S6 positive cells. These data seem to suggest the molecules are each upregulated/activated in different fibres or at different timepoints. The data do not seem to support the suggestion that mTORC1 is activated downstream of JUN.

Thank you! We agree, and we have now removed inferences to which of these pathways is activated first since our experiments do not allows us to reach a firm conclusion. Our hypothesis would be that mTORC1 is activated earlier in the presence of subtler mitochondrial dysfunction, while high c-Jun expression happens later, following mitochondrial loss and preceding demyelination.

The final two sentences in the first paragraph of the Discussion might be more appropriately placed elsewhere, as (unless I have misunderstood their data), a protective role for the ISR is not explicitly shown in the manuscript.

Thank you for the suggestion. We decided for removing these two sentences since this is a topic explored in our previous manuscript.

Reviewer #3 (Recommendations for the authors):1) Figure 3, can the authors provide data regarding the efficiency of Phb1 knockout in SCs in their mouse line? Do some Phb1-deficient SCs maintain an intact mitochondrial network?

Thank you for your question! From our characterization of this mouse line, and previous characterization of the P0-Cre line, we believe that recombination in SCs nears 100% of the cells. This can be illustrated by mRNA analyses that show a severe reduction in Phb1 levels in Phb1-SCKO mice (Figure 1c in Della-Flora Nunes et al., 2021). However, although we do see a small reduction in Phb1 protein levels from sciatic nerve lysates (Supplementary Figure 1b in Della-Flora Nunes et al., 2021), immunostaining experiments with different anti-PHB1 antibodies failed to show a significant reduction in PHB1 in our hands. We are unsure if this is due to poor specificity of the commercial PHB1 antibodies or to the high stability of the prohibitin proteins (He et al., 2008). This gets further complicated by the fact that the entire mitochondrial network seems to get lost in about 20% of all myelinating SCs in the sciatic nerve of P40 Phb1-SCKO mice (Figure 3h inDella-Flora Nunes et al., 2021). Therefore, it is hard to say if Phb1-deficient SCs are able to maintain an intact mitochondrial network, or if the whole mitochondrial network is dismantled when PHB1 levels reach a critically low level.

2) Figure 4: Since these are teased nerve fibers, not adjacent sections, please describe the detailed methods for immunofluorescence detection of DAPI and protein targets (JUN, P0, MBP, p-S6).

Thank you for your comment! We have now included an expanded description of the immunofluorescence method used in our analysis.

3) Lines 332 – 335: Please provide a more detailed discussion of the differential changes in Gdnf, Shh, and Cdh1 versus Mbp and Mpz. Appropriate references should be provided here.

Based on this suggestion and comments by Reviewer #1, we have adjusted this sentence and added more information about these c-Jun targets, the evidence demonstrating that c-Jun modulates their expression and the specificity of these targets for this pathway. Please see also response to reviewer #1.

4) Lines 439 – 440, "deletion of Jun in Phb1-SCKO mice had a dose-dependent effect on levels of p-4E-BP1 and p-S6." While the effect on p-4E-BP1 does appear to be dose-dependent, there is not a dose-dependent effect on p-S6.

We have changed this sentence to “Deletion of *Jun* in Phb1-SCKO mice had a dose-dependent effect on levels of p-4E-BP1 and also reduced p-S6 when both *Jun* alleles were deleted”

5) In the discussion, it is important to acknowledge that although Phb1 is primarily a mitochondrial protein, it has also been localized to the cytosol and nucleus under certain conditions. The nuclear localization is of considerable relevance here as it may allow Phb1 to interact with transcription factors. In addition, there appear to be reports of Phb2 interacting with JNK to activate c-JUN. This point may need to be discussed.

Thank you for your suggestion, we have added this information to our discussion to the paragraph that now reads:

“… Even though we favor an indirect role of PHBs on the activation of the mTORC1/c-Jun axis, we cannot rule out a direct interaction, and PHB2 was found to be a putative mTORC1 interactor in human T lymphoblasts (CCRF-CEM) and human embryonic kidney (HEK293) cells (Rahman et al., 2014), while PHB1 was found to bind to the mTOR inhibitor FK506 binding protein 8 (FKBP8) in different cell lines (Zhang et al., 2020), to inhibit c-Jun N-terminal kinase (*JNK*) signaling in cancer cell lines (Yang et al., 2019) and to stimulate c-Jun expression in cells of the colon of a mouse model of colitis (Kathiria et al., 2013). It is worth noting that, although PHBs are mostly found in the mitochondria, they can be present in the cytosol and nucleus of some cells in specific conditions (Thuaud et al., 2013), which could allow them to directly interact with transcription factors and signaling cascades.”

References

Beirowski, B., Wong, K. M., Babetto, E. and Milbrandt, J. 2017. mTORC1 promotes proliferation of immature Schwann cells and myelin growth of differentiated Schwann cells. *Proc Natl Acad Sci U S A,* 114**,** E4261-e4270.

Chen, B., Tan, Z., Gao, J., Wu, W., Liu, L., Jin, W., Cao, Y., Zhao, S., Zhang, W., Qiu, Z., Liu, D., Mo, X. And Li, W. 2015. Hyperphosphorylation of ribosomal protein S6 predicts unfavorable clinical survival in non-small cell lung cancer. *J Exp Clin Cancer Res,* 34**,** 126.

Dastidar, S. G., Pham, M. T., Mitchell, M. B., Yeom, S. G., Jordan, S., Chang, A., Sopher, B. L. And La Spada, A. R. 2020. 4E-BP1 Protects Neurons from Misfolded Protein Stress and Parkinson's Disease Toxicity by Inducing the Mitochondrial Unfolded Protein Response. *J Neurosci,* 40**,** 8734-8745.

Della-Flora Nunes, G., Wilson, E. R., Marziali, L. N., Hurley, E., Silvestri, N., He, B., O’malley, B. W., Beirowski, B., Poitelon, Y., Wrabetz, L. And Feltri, M. L. 2021. Prohibitin 1 is essential to preserve mitochondria and myelin integrity in Schwann cells. *Nature Communications,* 12.

Denarier, E., Forghani, R., Farhadi, H. F., Dib, S., Dionne, N., Friedman, H. C., Lepage, P., Hudson, T. J., Drouin, R. And Peterson, A. 2005. Functional organization of a Schwann cell enhancer. *J Neurosci,* 25**,** 11210-7.

Fontana, X., Hristova, M., Da Costa, C., Patodia, S., Thei, L., Makwana, M., Spencer-Dene, B., Latouche, M., Mirsky, R., Jessen, K. R., Klein, R., Raivich, G. And Behrens, A. 2012. c-Jun in Schwann cells promotes axonal regeneration and motoneuron survival via paracrine signaling. *J Cell Biol,* 198**,** 127-41.

Hagner, P. R., Mazan-Mamczarz, K., Dai, B., Balzer, E. M., Corl, S., Martin, S. S., Zhao, X. F. And Gartenhaus, R. B. 2011. Ribosomal protein S6 is highly expressed in non-Hodgkin lymphoma and associates with mRNA containing a 5' terminal oligopyrimidine tract. *Oncogene,* 30**,** 1531-41.

Harty, B. L., Coelho, F., Pease-Raissi, S. E., Mogha, A., Ackerman, S. D., Herbert, A. L., Gereau, R. W., Golden, J. P., Lyons, D. A., Chan, J. R. And Monk, K. R. 2019. Myelinating Schwann cells ensheath multiple axons in the absence of E3 ligase component Fbxw7. *Nat Commun,* 10**,** 2976.

He, B., Feng, Q., Mukherjee, A., Lonard, D. M., Demayo, F. J., Katzenellenbogen, B. S., Lydon, J. P. And O'malley, B. W. 2008. A repressive role for prohibitin in estrogen signaling. *Mol Endocrinol,* 22**,** 344-60.

Hung, H. A., Sun, G., Keles, S. And Svaren, J. 2015. Dynamic regulation of Schwann cell enhancers after peripheral nerve injury. *J Biol Chem,* 290**,** 6937-50.

Leblanc, S. E., Jang, S. W., Ward, R. M., Wrabetz, L. And Svaren, J. 2006. Direct regulation of myelin protein zero expression by the Egr2 transactivator. *J Biol Chem,* 281**,** 5453-60.

Musa, J., Orth, M. F., Dallmayer, M., Baldauf, M., Pardo, C., Rotblat, B., Kirchner, T., Leprivier, G. And Grünewald, T. G. 2016. Eukaryotic initiation factor 4E-binding protein 1 (4E-BP1): a master regulator of mRNA translation involved in tumorigenesis. *Oncogene,* 35**,** 4675-88.

Norrmen, C., Figlia, G., Pfistner, P., Pereira, J. A., Bachofner, S. And Suter, U. 2018. mTORC1 Is Transiently Reactivated in Injured Nerves to Promote c-Jun Elevation and Schwann Cell Dedifferentiation. *J Neurosci,* 38**,** 4811-4828.

Parkinson, D., Bhaskaran, A., Arthur-Farraj, P., Noon, L., Woodhoo, A., Lloyd, A., Feltri, M., Wrabetz, L., Behrens, A., Mirsky, R. And Jessen, K. 2008. c-Jun is a negative regulator of myelination. *The Journal of cell biology,* 181.

Sheean, M. E., Mcshane, E., Cheret, C., Walcher, J., Müller, T., Wulf-Goldenberg, A., Hoelper, S., Garratt, A. N., Krüger, M., Rajewsky, K., Meijer, D., Birchmeier, W., Lewin, G. R., Selbach, M. And Birchmeier, C. 2014. Activation of MAPK overrides the termination of myelin growth and replaces Nrg1/ErbB3 signals during Schwann cell development and myelination. *Genes Dev,* 28**,** 290-303.

Sherman, D., Krols, M., Wu, L., Grove, M., Nave, K., Gangloff, Y. And Brophy, P. 2012. Arrest of myelination and reduced axon growth when Schwann cells lack mTOR. *The Journal of neuroscience,* 32.

Yamaguchi, S., Ishihara, H., Yamada, T., Tamura, A., Usui, M., Tominaga, R., Munakata, Y., Satake, C., Katagiri, H., Tashiro, F., Aburatani, H., Tsukiyama-Kohara, K., Miyazaki, J., Sonenberg, N. And Oka, Y. 2008. ATF4-mediated induction of 4E-BP1 contributes to pancreatic β cell survival under endoplasmic reticulum stress. *Cell Metab,* 7**,** 269-76.